# Streaming Sliced Optimal Transport

**Khai Nguyen** [1]

## Abstract

Sliced optimal transport (SOT), or sliced Wasserstein (SW) distance, is widely recognized for its statistical and computational scalability. In this work, we further enhance computational scalability by proposing the first method for estimating SW from sample streams, called *streaming sliced Wasserstein* (Stream-SW). To define Stream-SW, we first introduce a streaming estimator of the one-dimensional Wasserstein distance (1DW). Since the 1DW has a closed-form expression, given by the integral of the absolute difference between the quantile functions of the compared distributions, we leverage quantile approximation techniques for sample streams to define a streaming 1DW estimator. By applying the streaming 1DW to all projections, we obtain Stream-SW. The key advantage of Stream-SW is its low memory complexity while providing theoretical guarantees on the approximation error. We demonstrate that Stream-SW achieves a more accurate approximation of SW than random subsampling, with lower memory consumption, when comparing Gaussian distributions and mixtures of Gaussians from streaming samples. Additionally, we conduct experiments on point cloud classification, point cloud gradient flows, and streaming change point detection to further highlight the favorable performance of the proposed Stream-SW.

## 1. Introduction

Optimal transport and the Wasserstein distance (Gabriel & Marco, 2019; Villani, 2008) have led to numerous advancements in machine learning, statistics, and data science. For example, the Wasserstein distance has been used to enhance generative models such as generative adversarial networks

Code for this paper is published at `https://github.com/khainb/StreamSW`. [1]Department of Statistics and Data Sciences, University of Texas at Austin, USA. Correspondence to: Khai Nguyen <khainb@utexas.edu>.

*Proceedings of the $43^{rd}$ International Conference on Machine Learning*, Seoul, South Korea. PMLR 306, 2026. Copyright 2026 by the author(s).

(GANs) (Arjovsky et al., 2017), autoencoders (Tolstikhin et al., 2018), and diffusion models via flow matching (Lipman et al., 2023; Pooladian et al., 2023; Tong et al., 2024). Additionally, it serves as a distortion (reconstruction) metric in point cloud applications (Achlioptas et al., 2018), and aids in approximate Bayesian inference (Bernton et al., 2019; Ambrogioni et al., 2018; Srivastava et al., 2018; Nguyen et al., 2026), among other applications.

Despite these benefits, the Wasserstein distance is computationally expensive. Specifically, its time complexity scales as $\mathcal{O}(n^3 \log n)$ (Pele & Werman, 2009), where $n$ is the maximum number of support points across two distributions. Moreover, it suffers from the curse of dimensionality, with a sample complexity of order $\mathcal{O}(n^{-1/d})$ (Fournier & Guillin, 2015), where $d$ is the number of dimensions. Consequently, in high-dimensional settings, a significantly larger number of samples is required to accurately approximate the distance between two unknown distributions using their empirical counterparts. This limitation makes the Wasserstein distance both statistically and computationally impractical.

Along with entropic regularization (Cuturi, 2013), which can reduce the time complexity of computing the Wasserstein distance to $\mathcal{O}(n^2)$ and overcome the curse of dimensionality, sliced optimal transport (Nguyen, 2025) (SOT) or the sliced Wasserstein (SW) distance (Rabin et al., 2010) provides an alternative approach to mitigate computational challenges. SW is defined as the expectation of the one-dimensional Wasserstein (1DW) distance between random one-dimensional Radon projections (Helgason, 2011) of two original distributions. Since 1DW has a closed-form solution, given by the absolute difference between the quantile functions, SW achieves a time complexity of $\mathcal{O}(n \log n)$ and a memory complexity of $\mathcal{O}(n)$ when comparing discrete distributions with at most $n$ supports. More importantly, SW distance does not suffer from the curse of dimensionality, with a sample complexity of $\mathcal{O}(n^{-1/2})$, making it both computationally and statistically scalable in any dimension. As a result, SW distance has been successfully applied in various domains, including generative models (Deshpande et al., 2018), domain adaptation (Lee et al., 2019), clustering (Kolouri et al., 2018), gradient flows (Liutkus et al., 2019; Bonet et al., 2022), Bayesian inference computation (Nadjahi et al., 2020a; Yi & Liu, 2021), posterior summarization (Nguyen & Mueller, 2026), two-sample test-

ing (Hu & Lin, 2025), improving OT estimations (Truong & Nguyen, 2026; Nguyen, 2026) and more.

Streaming data is increasingly common in scientific and technological applications, especially with small devices such as Internet of Things (IoT) devices, which collect, process, and transmit large volumes of data in real time. These devices face challenges due to limited memory and computational power, requiring streaming algorithms that process each data point independently and efficiently, without storing large amounts of data. This drives the need for single-pass algorithms that use minimal memory. As distribution comparison is one of the most vital tasks in data science, computing the distance between two distributions from their sample streams becomes a natural and important challenge to address from theory to practice.

Recently, there has been growing interest in computing optimal transport, such as the Wasserstein distance, from streaming samples of two distributions. Online Sinkhorn (Mensch & Peyré, 2020) was the first algorithm to address entropic optimal transport in the streaming/online setting. However, Online Sinkhorn has a time complexity of $\mathcal{O}(n^2)$ and a memory complexity of $\mathcal{O}(n)$, as it requires storing all previously visited data. As a result, it remains impractical for streaming applications. Compressed Online Sinkhorn (Wang et al., 2023) attempts to mitigate this issue by employing measure compression. However, performing measure compression, such as Gaussian quadrature, is computationally expensive, exhibiting super-cubic complexity, which is undesirable in streaming settings with limited computational resources. Furthermore, the compression in Compressed Online Sinkhorn scales poorly with dimension, with a compression rate of $\mathcal{O}(m^{-1/d})$, where $m$ is the compressed support size. Consequently, the algorithm also remains computationally expensive. A natural question arises: "Could we adapt SOT and SW to the streaming setting?"

We provide the first answer to the question by proposing the first streaming version of the SW distance which can handle streaming samples from distributions in limited memory settings. To achieve this goal, we bridge the literature on *streaming quantile approximation* and *sliced optimal transport* by leveraging the previously discussed insight that the 1DW is computed based on quantile functions. In summary, our contributions are three-fold:

1. We first propose using quantile sketches as a data structure for estimating cumulative distribution functions (CDFs) and quantile functions of distributions from streaming samples. We then derive probabilistic bounds that characterize the approximation guarantees of these streaming estimators for both CDFs and quantile functions. Leveraging these results, we introduce the first streaming estimators of the one-dimensional Wasserstein (1DW) distance and the corresponding one-dimensional optimal transport map from sam-

ple streams. In addition, we establish probabilistic bounds for the approximation errors of the proposed streaming estimators for discussed quantities.

2. Using the streaming estimator of 1DW for one-dimensional projections, we introduce streaming Sliced Wasserstein (Stream-SW), the first streaming estimator for the SW distance. We then extend the probabilistic bounds for the approximation errors from 1DW to Stream-SW. Moreover, we discuss how Stream-SW can be estimated using a finite number of projections and derive the corresponding guarantees for the proposed estimator. Furthermore, we analyze the computational complexity of Stream-SW, including both space and time complexities with respect to the number of streaming samples and the size of the quantile sketches. In summary, the space complexity of Stream-SW is logarithmic in the number of streaming samples and linear in the initial sketch size, while the time complexity is nearly linear in the number of streaming samples and nearly linear in the initial sketch size.

3. Empirically, we demonstrate that Stream-SW achieves a more accurate approximation and offers better downstream performance compared to SW with random sampling (i.e., retaining only a random subset of streaming samples). Specifically, we first conduct simulations comparing Gaussians and Gaussian mixtures from sample streams. We then apply Stream-SW to streaming point-cloud classification using K-nearest neighbors, followed by a comparison in a streaming gradient flow setting. Finally, we showcase the effectiveness of Stream-SW in streaming change point detection using Kinect gesture dataset.

**Organization.** We begin by reviewing background on the sliced Wasserstein distance in Section 2. Next, we discuss the definition of quantile sketch, propose streaming CDFs estimation, streaming quantile function estimation, streaming 1DW, Stream-SW, and their theoretical and computational properties in Section 3. Section 4 presents simulations and experiments. Finally, we conclude in Section 5. Additional materials, including technical proofs, detailed experiment descriptions, and supplementary experimental results, are provided in the Appendices.

**Notations.** For any $d \geq 2$, we define the unit hypersphere as $\mathbb{S}^{d-1} := \{\theta \in \mathbb{R}^d \mid \|\theta\|_2^2 = 1\}$ and denote $\mathcal{U}(\mathbb{S}^{d-1})$ as the uniform distribution over it. The set of all probability measures on a given set $\mathcal{X}$ is represented by $\mathcal{P}(\mathcal{X})$. For $p \geq 1$, we denote by $\mathcal{P}_p(\mathcal{X})$ the collection of probability measures on $\mathcal{X}$ that possess finite $p$-moments. For two sequences $a_n$ and $b_n$, the notation $a_n = \mathcal{O}(b_n)$ signifies that there exists a universal constant $C$ such that $a_n \leq Cb_n$ for all $n \geq 1$. The pushforward measure of $\mu$ under a function $f : \mathbb{R}^d \to \mathbb{R}$ defined as $f(x) = \theta^\top x$ is denoted by $\theta\sharp\mu$. For a vector $X \in \mathbb{R}^{dm}$, where $X = (x_1, \ldots, x_m)$, the empirical measure associated with $X$ is given by $P_X := \frac{1}{m}\sum_{i=1}^{m}\delta_{x_i}$.

## 2. Preliminaries

We first review definitions and computational properties of Wasserstein distance and SW distance.

**Wasserstein distance.** For the order $p \geq 1$, the Wasserstein-$p$ distance (Villani, 2008; Gabriel & Marco, 2019) between two distributions $\mu \in \mathcal{P}_p(\mathbb{R}^d)$ and $\nu \in \mathcal{P}_p(\mathbb{R}^d)$ (dimension $d \geq 1$) is defined as:

$$W_p^p(\mu,\nu) = \inf_{\pi \in \Pi(\mu,\nu)} \int_{\mathcal{X} \times \mathcal{Y}} \|x-y\|_p^p d\pi(x,y), \quad (1)$$

where $\Pi(\mu,\nu)$ is the set of all transportation plans i.e., joint distributions which have marginals be $\mu$ and $\nu$ respectively. When $\mu$ and $\nu$ are discrete distributions i.e., $\mu = \sum_{i=1}^n \alpha_i \delta_{x_i}$ ($n \geq 1$) and $\nu = \sum_{j=1}^m \beta_j \delta_{y_j}$ ($m \geq 1$) where $\sum_{i=1}^n \alpha_i = \sum_{j=1}^m \beta_j = 1$ and $\alpha_i \geq 0, \beta_j \geq 0$ for all $i = 1, \ldots, n$ and $j = 1, \ldots, m$, Wasserstein distance between $\mu$ and $\nu$ is defined as follows:

$$W_p^p(\mu,\nu) = \min_{\gamma \in \Gamma(\alpha,\beta)} \sum_{i=1}^n \sum_{j=1}^m \|x_i - y_j\|_p^p \gamma_{ij}, \quad (2)$$

where $\Gamma(\alpha,\beta) = \{\gamma \in \mathbb{R}_+^{n \times m} \mid \gamma \mathbf{1} = \alpha, \gamma^\top \mathbf{1} = \beta\}$. Without losing generality, we assume that $n \geq m$. Therefore, the time complexity for solving this linear programming is $\mathcal{O}(n^3 \log n)$ (Gabriel & Marco, 2019) and $\mathcal{O}(n^2)$ in turn which are very expensive. Moreover, from (Fournier & Guillin, 2015), we have the following sample complexity: $\mathbb{E}\left[|W_p(\mu_n, \nu_m) - W_p(\mu,\nu)|\right] = \mathcal{O}(\min\{n,m\}^{-1/d})$. Here $\mu_n$ and $\nu_m$ are empirical distributions over $n$ and $m$ i.i.d samples from $\mu$ and $\nu$ in turn. The above result implies that we need $\min\{n,m\}$ to be large in high-dimension in order to reduce the approximation error. However, as discussed, Wasserstein scales poorly with $n$ and $m$. Therefore, sliced Wasserstein (Rabin et al., 2014; Bonneel et al., 2015) (SW) is proposed as an alternative solution. The key idea of SW is to utilize the closed-form solution of optimal transport in one-dimension.

**One-dimensional Optimal Transport.** Given $\mu \in \mathcal{P}(\mathbb{R})$ and $\nu \in \mathcal{P}(\mathbb{R})$, we have the optimal transport mapping from $\mu$ to $\nu$ is $F_\nu^{-1} \circ F_\mu$ (Bonnotte, 2013; Gabriel & Marco, 2019), where $F_\mu$ and $F_\nu$ are CDF of $\mu$ and $\nu$ respectively. The one-dimensional Wasserstein distance between $\mu$ and $\nu$ is then defined as follows:

$$W_p^p(\mu,\nu) = \int_{\mathbb{R}} \left|x - F_\nu^{-1}(F_\mu(x))\right|^p d\mu(x)$$
$$= \int_0^1 \left|F_\mu^{-1}(q) - F_\nu^{-1}(q)\right|^p dq, \quad (3)$$

When $\mu$ and $\nu$ are discrete distributions i.e., $\mu = \sum_{i=1}^n \alpha_i \delta_{x_i}$ ($n \geq 1$) and $\nu = \sum_{j=1}^m \beta_j \delta_{y_j}$ ($m \geq 1$), quan-

tile functions of $\mu$ and $\nu$ can be written as follows:

$$F_\mu^{-1}(q) = \sum_{i=1}^n x_{(i)} I \left( \sum_{j=1}^{i-1} \alpha_{(j)} < q \leq \sum_{j=1}^i \alpha_{(j)} \right),$$
$$F_\nu^{-1}(q) = \sum_{j=1}^m y_{(j)} I \left( \sum_{i=1}^{j-1} \beta_{(i)} < q \leq \sum_{i=1}^j \beta_{(i)} \right),$$

where $x_{(1)} \leq \ldots \leq x_{(n)}$ and $y_{(1)} \leq \ldots \leq y_{(m)}$ are order statistics. This construction of quantile functions are equivalent to northwest corner algorithm for discrete optimal transport (Gabriel & Marco, 2019). Therefore, computing one-dimensional Wasserstein distance only cost $\mathcal{O}(n \log n)$ and $\mathcal{O}(n)$ in time complexity and space complexity respectively (assuming $m \leq n$).

**Sliced Wasserstein Distance.** For $p \geq 1$, the *Sliced Wasserstein (SW)* distance (Bonneel et al., 2015) of $p$-th order between two distributions $\mu \in \mathcal{P}(\mathbb{R}^d)$ and $\nu \in \mathcal{P}(\mathbb{R}^d)$ is defined as:

$$SW_p^p(\mu,\nu) = \mathbb{E}_{\theta \sim \mathcal{U}(\mathbb{S}^{d-1})}[W_p^p(\theta \sharp \mu, \theta \sharp \nu)], \quad (4)$$

where $\theta \sharp \mu$ and $\theta \sharp \nu$ are the one-dimensional push-forward distributions by applying Radon Transform (RT) (Helgason, 2011) on the probability density functions (pdfs) of $\mu$ and $\nu$ with the projecting direction $\theta$ i.e., $\mathcal{R}_\theta(x) = \theta^\top x$. Using the closed-form of 1DW, we can further rewrite:

$$SW_p^p(\mu,\nu)$$
$$= \mathbb{E}_{\theta \sim \mathcal{U}(\mathbb{S}^{d-1})} \left[ \int_0^1 \left|F_{\theta \sharp \mu}^{-1}(q) - F_{\theta \sharp \nu}^{-1}(q)\right|^p dq \right], \quad (5)$$

which boils down to the expectation of difference between two quantile functions of projected distributions. This observation is the key insight in design later streaming estimator of SW (Stream-SW).

Due to the intractability of the expectation in (4), numerical approximation need to be used (Nguyen et al., 2024; Leluc et al., 2024; Sisouk et al., 2025). For example, Monte Carlo estimation with $\theta_1, \ldots, \theta_L \overset{i.i.d}{\sim} \mathcal{U}(\mathbb{S}^{d-1})$ ($L \geq 1$) is often used:

$$\widehat{SW}_p^p(\mu,\nu;L) = \frac{1}{L} \sum_{l=1}^L W_p^p(\theta_l \sharp \mu, \theta_l \sharp \nu), \quad (6)$$

where $L$ is often referred to as the number of projections. The time complexity and space complexity of SW are $\mathcal{O}(Ln \log n + Ldn)$ and $\mathcal{O}(Ld + nd)$ in turn. Moreover, from (Nadjahi et al., 2020b; Manole et al., 2022; Nietert et al., 2022; Boedihardjo, 2025), we have that $\mathbb{E}\left[|SW_p(\mu_n, \nu_m) - SW_p(\mu,\nu)|\right] = \mathcal{O}(\min\{n,m\}^{-1/2})$. Therefore, SW is statistically scalable and computationally scalable in both the number of supports and the number of dimensions.

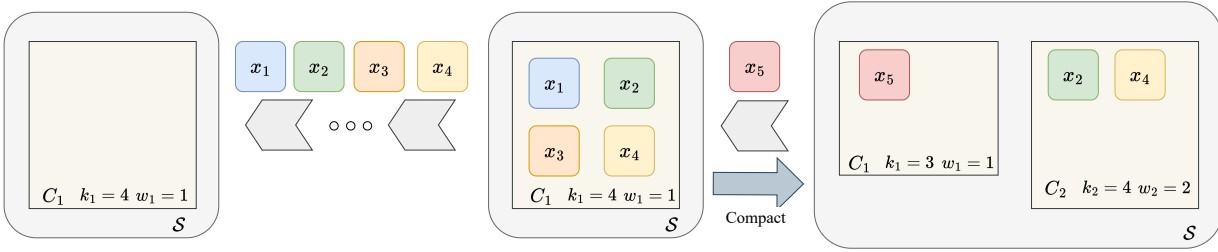

*Figure 1.* The figure shows the compacting process of a KLL sketch.

# 3. Streaming Sliced Optimal Transport

In this section, we discuss computing sliced optimal transport from sample streams. In greater detail, we receive samples $x_1, \ldots, x_n$ ($n > 1$) and $y_1, \ldots, y_m$ ($m > 1$) in a streaming fashion in arbitrary order from two interested $d > 1$ dimensional distributions $\mu$ and $\nu$, respectively. The goal is to form an approximation of the sliced optimal transport between $\mu$ and $\nu$ with a *budget-constrained* memory. The constraint comes from the nature of the streaming setting where storing all samples is impractical since $n$ and $m$ are often very large and might be unknown.

## 3.1. Quantile Sketch

As discussed in Section 2, the key computation of one-dimensional optimal transport and sliced optimal transport (SOT) rely on evaluating CDFs and quantile functions. To enable computing SOT from sample streams, it is a must to compute CDFs and quantile functions from sample streams. The key technique for such task is to build a data structure, named *quantile sketch*, which can yield an approximate number for any quantile query input.

**Quantile Sketch.** Quantile sketch $\mathcal{S}$ is a data structure that store a subset of samples from the data streams of $n$ samples (Greenwald & Khanna, 2001). In this work, we focus on using the KLL sketch (Karnin et al., 2016) which is the optimal randomized comparison-based quantile sketch. In addition, KLL sketch is also mergeable which allows distributed computation. The key component of an KLL sketch is a *compactor* $C$ which is a set of $k$ elements with an associated weight $w$ i.e., $C = \{x_1, \ldots, x_k\}$. A compactor, as its name, can compact its $k$ elements into $k/2$ elements of weight $2w$. In greater detail, the items are first sorted. Then, either the even or the odd elements in the set are chosen. For the KLL sketch, there are $H$ hierarchical compactors indexed by their height i.e., $C_h$ for $h = 1, \ldots, H$. The associated weight of $C_h$ is $w_h = 2^{h-1}$. For the capacity of $C_h$, we set $k_h = [k(c)^{H-h}] + 1$ ($c = 2/3$) for a given initial capacity $k < n$. When the compactor $C_h$ does compaction (triggered when it reaches its capacity), the chosen elements are put into the compactor $C_{h+1}$. It is worth noting that if $k_h$ is odd, the compaction is conducted only on $k - 1$ first items after sorting to make sure that the total weights is still

$n$. An example of the compacting process of a KLL sketch in shown in Figure 1.

**Complexities of KLL sketch.** From Karnin et al. (2016), given a stream of $n > k$ samples, we have $n \geq k_{H-1} w_{H-1} = k_{H-1} 2^{H-2}$ which gives $H \leq \log(n/k_{H-1}) + 2 \leq \log((3n)/(2k)) + 2$. The total capacity of all compactors is $\sum_{h=1}^{H} k_h \leq \sum_{h=1}^{H} (k(2/3)^{H-h} + 2) \leq 3k + 2H \leq 3k + 2(\log((3n)/(2k)) + 2)$ which yields the space complexity of $\mathcal{O}(k + \log(n/k))$. Moreover, the number of compact operations at height $h$ satisfies $m_h \leq n/(k_h w_h)$, so the total number of items processed by compactions is $\sum_{h=1}^{H} k_h m_h \leq \sum_{h=1}^{H} \frac{n}{w_h} = n \sum_{h=1}^{H} 2^{-(h-1)} \leq 2n$, i.e. each stream item is touched a constant number of times as weights double up the hierarchy. Including the sort performed at each compaction, the total compaction time is $\mathcal{O}(n \log k)$.

## 3.2. Streaming CDF and Quantile Function Estimation

Given the quantile sketch, we can form a discrete distribution from samples and their weights from the sketch. The discrete distribution is $\frac{1}{n} \sum_{h=1}^{H} \sum_{x \in C_h} w_h \delta_x$. With this distribution, we can form streaming estimations of CDF and quantile function.

**Streaming CDF estimation.** We form the following estimation of CDF from a sketch $\mathcal{S}$: $F_{\mathcal{S}}(y) = \frac{1}{n} \sum_{h=1}^{H} \sum_{x \in C_h} w_h I(x \leq y)$. We now discuss the approximation guarantee of $F_{\mathcal{S}}(y)$ with respect to the empirical CDF: $F_n(y) = \frac{1}{n} \sum_{i=1}^{n} I(x_i \leq y)$.

**Proposition 1.** *Given $\mathcal{S}_{\mu,k}$ be the quantile sketch with initial size $k > 0$ constructed from streaming samples $x_1, \ldots, x_n$ from $\mu \in \mathcal{P}(\mathbb{R})$, the following probability bound holds:*

$$\mathbb{P}\left(|F_{\mathcal{S}_{\mu,k}}(x) - F_n(x)| > \epsilon\right) \leq 2 \exp\left(-Ck^2\epsilon^2\right), \quad (7)$$

*$\forall x \in \mathbb{R}$, where $C > 0$ is a constant. When $\mu$ has compact support with diameter $R > 0$, the following probability bound holds:*

$$\mathbb{P}\left(\int_{\mathbb{R}} |F_{\mathcal{S}_{\mu,k}}(x) - F_n(x)| \mathrm{d}x > \epsilon\right)$$
$$\leq \frac{A_R}{\epsilon} \exp\left(-C_R k^2 \epsilon^2\right), \quad (8)$$

where $A_R, C_R > 0$ are constants which depend on $R$.

The proof of Proposition 1 is given in Appendix A.1. Setting $(A_R/\epsilon)\exp(-C_R k^2 \epsilon^2) = \delta$, we need $k = \mathcal{O}\big((1/\epsilon)\sqrt{\log(1/(\epsilon\delta))}\big)$ to have $\epsilon$ error, uniformly over all query points, with failure probability $\delta$.

**Streaming quantile function estimation.** Similar to the case of CDF, we can also can form an estimation of the quantile function: $F_{\mathcal{S}_{\mu,k}}^{-1}(q) = \inf\{x : F_{\mathcal{S}_{\mu,k}}(x) \geq q\}$. We again can guarantee the quality of the approximation to the empirical quantile function $F_n^{-1}(q) = \inf\{x : F_n(x) \geq q\}$ as follows:

**Proposition 2.** *Given $\mathcal{S}$ be the quantile sketch with initial size $k > 1$ constructed from streaming samples $x_1, \ldots, x_n$ from $\mu \in \mathcal{P}(\mathbb{R})$, the following probability bound holds:*

$$\mathbb{P}\left(\left|F_{\mathcal{S}_{\mu,k}}^{-1}(q) - F_n^{-1}(q)\right| > \epsilon\right)$$
$$\leq 2\exp\left(-Ck^2\gamma_{q,\epsilon}^2\right), \qquad (9)$$

$\forall q \in [0,1]$, *where $C > 0$ is a constant and $\gamma_{q,\epsilon} = \min\big\{q - F_n(F_n^{-1}(q) - \epsilon), F_n(F_n^{-1}(q) + \epsilon) - q\big\}$. When $\mu$ has compact support with diameter $R > 0$, the following probability bound holds:*

$$\mathbb{P}\left(\int_0^1 |F_{\mathcal{S}_{\mu,k}}^{-1}(q) - F_n^{-1}(q)|\mathrm{d}q > \epsilon\right)$$
$$\leq \frac{A_R}{\epsilon}\exp\left(-C_R k^2 \epsilon^2\right), \qquad (10)$$

*where $A_R, C_R > 0$ are constants which depend on $R$.*

The proof of Proposition 2 is given in Appendix A.2. From the compact-support case, we also need $k = \mathcal{O}\big((1/\epsilon)\sqrt{\log(1/(\epsilon\delta))}\big)$ to have $\epsilon$ error with failure probability $\delta$.

**Streaming one-dimensional Wasserstein distance.** With the discussed streaming estimator for quantile functions, we are able to define a streaming estimator for 1DW. We are given $x_1, \ldots, x_n$ $(n > 1)$ and $y_1, \ldots, y_m$ $(m > 1)$ in a streaming fashion and in arbitrary order from two interested one-dimensional distributions $\mu$ and $\nu$. Let $\mu_n$ and $\nu_m$ be the two corresponding empirical distributions i.e., $\mu_n = \frac{1}{n}\sum_{i=1}^n \delta_{x_i}$ and $\nu_m = \frac{1}{m}\sum_{j=1}^m \delta_{y_j}$, from the sample streams, we build KLL sketches $\mathcal{S}_{\mu_n,k_1}$ and $\mathcal{S}_{\nu_m,k_2}$. After that, we can form the approximation of the one-dimensional Wasserstein distance between $\mu_n$ and $\nu_m$ as follows:.

**Definition 1.** Given two empirical distributions $\mu_n$ and $\nu_m$ observed in a streaming fashion, and their corresponding quantile sketches $\mathcal{S}_{\mu_n,k_1}$ and $\mathcal{S}_{\nu_m,k_2}$ with $k_1 > 1$ and $k_2 > 1$, streaming one-dimensional Wasserstein is defined as follow:

$$\widetilde{W}_p^p(\mu_n, \nu_m; \mathcal{S}_{\mu_n,k_1}, \mathcal{S}_{\nu_m,k_2})$$
$$= \int_0^1 \left|F_{\mathcal{S}_{\mu_n,k_1}}^{-1}(q) - F_{\mathcal{S}_{\nu_m,k_2}}^{-1}(q)\right|^p \mathrm{d}q. \qquad (11)$$

We can rewrite streaming one-dimensional Wasserstein as: $\widetilde{W}_p^p(\mu_n, \nu_m; \mathcal{S}_{\mu_n,k_1}, \mathcal{S}_{\nu_m,k_2}) = W_p^p\big(\mu_{\mathcal{S}_{\mu_n,k_1}}, \nu_{\mathcal{S}_{\nu_m,k_2}}\big)$, where $\mu_{\mathcal{S}} = \frac{1}{n}\sum_{h=1}^H \sum_{x \in C_h} w_h \delta_x$ denotes the discrete distribution over the quantile sketch $\mathcal{S}$. In addition, we can also define one-sided streaming one-dimensional Wasserstein by using only a quantile sketch for one distribution. Next, we discuss approximation guarantee of the proposed streaming estimator.

**Proposition 3.** *Let $\mu, \nu \in \mathcal{P}(\mathbb{R})$ be probability measures supported on an interval of diameter $R > 0$. Let $\mu_n$ and $\nu_m$ denote the empirical measures formed from $n$ and $m$ i.i.d. samples from $\mu$ and $\nu$, respectively. Let $\mathcal{S}_{\mu_n,k_1}$ and $\mathcal{S}_{\nu_m,k_2}$ be quantile sketches of initial sizes $k_1, k_2 > 0$ constructed from the samples defining $\mu_n$ and $\nu_m$. For any $p \geq 1$ and $\epsilon > 0$, the following bounds hold:*

$$\mathbb{P}\Big(\big|\widetilde{W}_p^p(\mu_n, \nu_m; \mathcal{S}_{\mu_n,k_1}, \mathcal{S}_{\nu_m,k_2}) - W_p^p(\mu_n, \nu_m)\big| > \epsilon\Big)$$
$$\leq \frac{A_{p,R}}{\epsilon}\exp\big(-C_{p,R}\min\{k_1^2, k_2^2\}\epsilon^2\big), \qquad (12)$$

*where $A_{p,R}, C_{p,R} > 0$ are constants depending only on $p$ and $R$, and*

$$\mathbb{P}\Big(\big|\widetilde{W}_p^p(\mu_n, \nu_m; \mathcal{S}_{\mu_n,k_1}, \mathcal{S}_{\nu_m,k_2}) - W_p^p(\mu, \nu)\big| > \epsilon\Big)$$
$$\leq \Big(\frac{a_{p,R}}{\epsilon} + 8\Big)\exp\big(-c_{p,R}\min\{k_1^2, k_2^2, n, m\}\epsilon^2\big), \quad (13)$$

*where $a_{p,R}, c_{p,R} > 0$ are constants depending on $p$ and $R$.*

The proof of Proposition 3 is given in Appendix A.3.

**Streaming one-dimensional transport map.** Let $\mathcal{S}_{\mu_n,k_1}$ and $\mathcal{S}_{\nu_m,k_2}$ be quantile sketches of sizes $k_1, k_2$ constructed from i.i.d. samples drawn from $\mu$ and $\nu$, respectively. Define the monotone transport map $T_{\mu\to\nu}(x) := F_\nu^{-1}(F_\mu(x))$, and its sketched approximation

$$\widetilde{T}(x) := F_{\mathcal{S}_{\nu_m,k_2}}^{-1}\big(F_{\mathcal{S}_{\mu_n,k_1}}(x)\big).$$

We can also guarantee the approximation of $\widetilde{T}(x)$ under an assumption about the continuity of $\nu$:

**Proposition 4.** *Let $\mu, \nu \in \mathcal{P}(\mathbb{R})$ be probability measures supported on an interval of diameter $R > 0$. Assume that $\nu$ is absolutely continuous with density $f_\nu$ satisfying $f_\nu(x) \geq a > 0$ for all $x$ in the support of $\nu$ (necessarily $a \leq 1/R$). Then for any $\epsilon > 0$,*

$$\mathbb{P}\Big(\int_{\mathbb{R}} \big|\widetilde{T}(x) - T_{\mu\to\nu}(x)\big|\,\mathrm{d}\mu(x) > \epsilon\Big)$$
$$\leq \Big(\frac{a_{a,R}}{\epsilon} + 8\Big)\exp\big(-c_{a,R}\min\{k_1^2, k_2^2, n, m\}\epsilon^2\big), \quad (14)$$

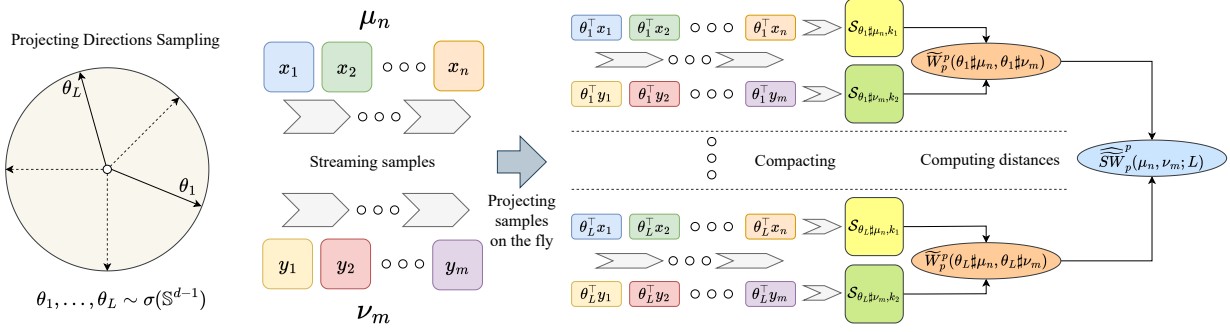

Figure 2. The figure shows the computational procedure of Stream-SW. In the figure, we denotes $\widetilde{W}_p^p(\mu_n, \nu_m; \mathcal{S}_{\mu_n,k_1}, \mathcal{S}_{\nu_m,k_2})$ as $\widetilde{W}_p^p(\mu_n, \nu_m)$ and $\widehat{\widetilde{SW}}_p^p(\mu_n, \nu_m; k_1, k_2, L)$ as $\widehat{\widetilde{SW}}_p^p(\mu_n, \nu_m; L)$.

where $a_{a,R}, c_{a,R} > 0$ *depend only on $a$ and $R$.*

The proof of Proposition 4 is given in Appendix A.4.

### 3.3. Streaming Sliced Wasserstein

With the previously discussed streaming estimator of quantile function, we can define Stream-SW as follows:

**Definition 2.** Let $k_1 > 1$, $k_2 > 1$, and $p \geq 1$. The *streaming Sliced Wasserstein* (Stream-SW) distance between two empirical distributions $\mu_n \in \mathcal{P}_p(\mathbb{R}^d)$ and $\nu_m \in \mathcal{P}_p(\mathbb{R}^d)$, whose supports are observed in a streaming manner, is defined by

$$\widetilde{SW}_p^p(\mu_n, \nu_m; k_1, k_2)$$
$$= \mathbb{E}_{\theta \sim \sigma}\left[\widetilde{W}_p^p\left(\theta\sharp\mu_n, \theta\sharp\nu_m; S_{\theta\sharp\mu_n,k_1}, S_{\theta\sharp\nu_m,k_2}\right)\right], \quad (15)$$

where $\sigma \in \mathcal{P}(\mathbb{S}^{d-1})$ is a slicing distribution on the unit sphere and $\widetilde{W}_p^p(\theta\sharp\mu_n, \theta\sharp\nu_m; S_{\theta\sharp\mu_n,k_1}, S_{\theta\sharp\nu_m,k_2})$ is defined in Definition 1.

From the failure probability of streaming quantile estimator in Proposition 3, we obtain similar result for Stream-SW.

**Corollary 1.** *Let $\mu, \nu \in \mathcal{P}_p(\mathbb{R}^d)$ be probability measures supported on a compact set of diameter $R > 0$. Let $\mu_n$ and $\nu_m$ be the empirical measures formed from $n$ and $m$ i.i.d. samples from $\mu$ and $\nu$, respectively. Let $k_1 > 1$, $k_2 > 1$, and let $S_{\theta\sharp\mu_n,k_1}$ and $S_{\theta\sharp\nu_m,k_2}$ be quantile sketches constructed from streaming observations of the projected empirical measures $\theta\sharp\mu_n$ and $\theta\sharp\nu_m$, where $\theta \in \mathbb{S}^{d-1}$. Then for any $p \geq 1$,*

$$\mathbb{E}\left[\left|\widetilde{SW}_p^p(\mu_n, \nu_m; k_1, k_2) - SW_p^p(\mu_n, \nu_m)\right|\right]$$
$$\leq C'_{p,R}\,\min\{k_1, k_2\}^{-1}, \quad (16)$$

*and*

$$\mathbb{E}\left[\left|\widetilde{SW}_p^p(\mu_n, \nu_m; k_1, k_2) - SW_p^p(\mu, \nu)\right|\right]$$
$$\leq c'_{p,R}\,\min\{k_1, k_2, \sqrt{n}, \sqrt{m}\}^{-1}, \quad (17)$$

*where $C'_{p,R} > 0$ and $c'_{p,R} > 0$ are constants depending only on $p$ and $R$.*

The proof of Corollary 1 is given in Appendix A.5.

**Numerical approximation.** As in the conventional SW, we need to approximate the expectation in Definition 2. There are some existing options such as Monte Carlo (MC) estimation (Bonneel et al., 2015) and quasi-Monte Carlo approximation and randomized quasi-Monte Carlo estimation (Nguyen et al., 2024). We refer the reader to a recent guide in (Sisouk et al., 2025) for a detailed discussion. By using one of the mentioned constructions, we can obtain a final set of projecting directions $\theta_1, \ldots, \theta_L \in \mathbb{S}^{d-1}$ with $L > 0$ which is referred to as the number of projections. We then can form the numerical approximation of streaming SW as follows:

$$\widehat{\widetilde{SW}}_p^p(\mu_n, \nu_m; k_1, k_2, L)$$
$$= \frac{1}{L}\sum_{l=1}^{L}\widetilde{W}_p^p(\theta_l\sharp\mu_n, \theta_l\sharp\nu_m; S_{\theta_l\sharp\mu_n,k_1}, S_{\theta_l\sharp\nu_m,k_2}). \quad (18)$$

We present the computational process of Stream-SW in Figure 2. For simplicity, we use MC estimation in this work i.e., $\theta_1, \ldots, \theta_L \overset{i.i.d}{\sim} \sigma(\mathbb{S}^{d-1})$. We then discuss the final theoretical guarantee result for the MC estimation.

**Theorem 1.** *Let $\mu, \nu \in \mathcal{P}_p(\mathbb{R}^d)$ be probability measures supported on a compact set of diameter $R > 0$. Let $\mu_n$ and $\nu_m$ be the empirical measures formed from $n$ and $m$ i.i.d. samples from $\mu$ and $\nu$, respectively. Let $k_1 > 1$, $k_2 > 1$, and let $S_{\theta\sharp\mu_n,k_1}$ and $S_{\theta\sharp\nu_m,k_2}$ be quantile sketches constructed from streaming observations of the projected empirical measures $\theta\sharp\mu_n$ and $\theta\sharp\nu_m$. Then for any $p \geq 1$, the following bound holds:*

$$\mathbb{E}\left[\left|\widehat{\widetilde{SW}}_p^p(\mu_n, \nu_m; k_1, k_2, L) - SW_p^p(\mu, \nu)\right|\right]$$
$$\leq c'_{p,R}\left(\min\{k_1, k_2, \sqrt{n}, \sqrt{m}, \sqrt{L}\}^{-1}\right), \quad (19)$$

*where $c'_{p,R} > 0$ is a constant depending only on $p$ and $R$.*

The proof of Theorem 1 is given in Appendix A.6.

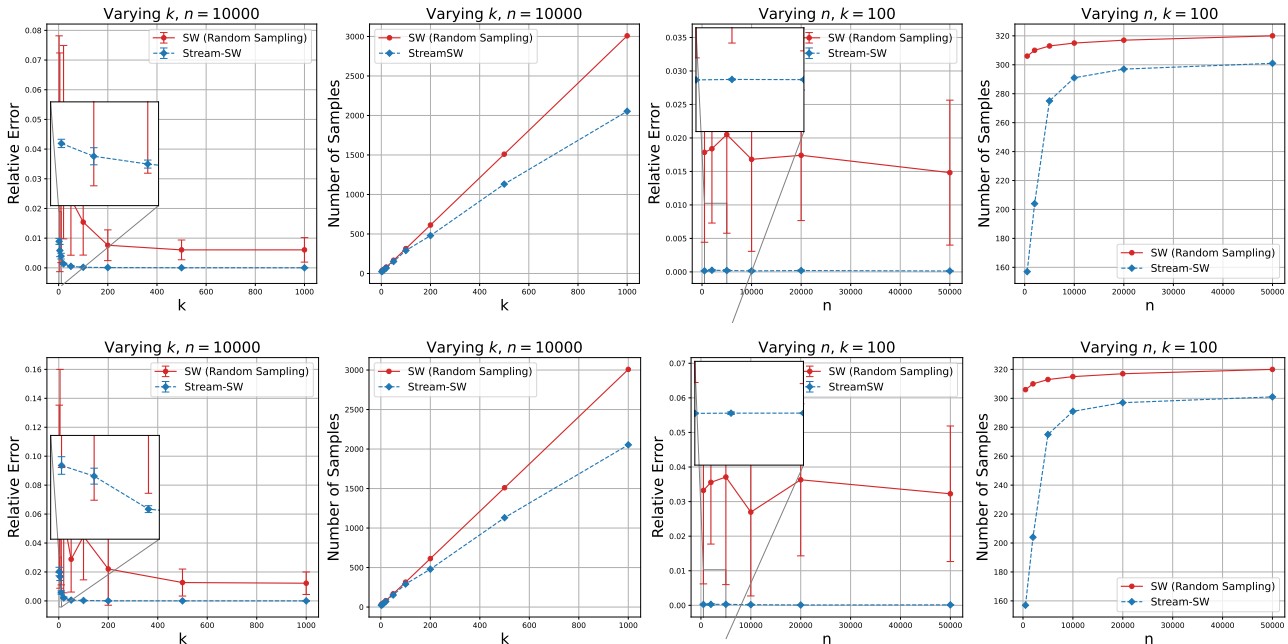

*Figure 3.* Relative errors ($|\cdot - SW_2^2(\mu_n, \nu_n)|$) and number of samples when comparing Gaussian distributions (first row) and mixture of Gaussians distributions (second row).

**Computational complexities.** Without losing generality, we assume that $k_1 \leq k_2 \leq k$, from the discussion of the space complexity of KLL sketches in Section 3.1, the space complexity of Stream-SW is $\mathcal{O}(L(k + \log(n/k)) + Ld)$ where $Ld$ is for storing projecting directions $\theta_1, \ldots, \theta_L$. The time complexity for compacting is $\mathcal{O}(Ln \log k)$. For computing 1DW, the time complexity is $\mathcal{O}(L(k + \log(n/k)) \log((k + \log(n/k))))$. Adding time complexity for projection $\mathcal{O}(Ldn)$, we have the final time complexity $\mathcal{O}(Ldn + Ln \log k + L(k + \log(n/k)) \log(k + \log(n/k)))$, dominated by the projection term whenever $d \gtrsim \log k$. In summary, the time complexity of Stream-SW is near linear in $n$ and depends on the initial sketch size $k$ near-linearly. The space complexity of Stream-SW is logarithmic in $n$ and linear in $k$. We recall that these complexities are theoretical and the practical computational time depends on the implementation.

# 4. Experiments

We first do approximation error analysis in Section 4.1. We then conduct experiments on streaming point-cloud classification in Section 4.2 and streaming gradient flow in Section 4.3. Finally, we discuss streaming change point detection in Section 4.4. Additional experiments such as computational time analysis are given in Appendix C.

## 4.1. Approximation Error Analysis

We first analyze the approximation error of Stream-SW in the non-compact support setting to complement our theoretical results for compactly supported distributions. We generate samples from the two Gaussian distributions $\mathcal{N}((-1,-1), I)$ and $\mathcal{N}((2,2), I)$. We investigate both the approximation error and the number of samples retained from the stream as the sketch size $k$ and the number of supports $n$ vary. Specifically, we fix the number of projections at $L = 1000$, vary $k \in 2, 5, 10, 20, 50, 100, 200, 500, 1000$ while keeping $n = 10000$, and vary $n \in 500, 2000, 5000, 10000, 20000, 50000$ while fixing $k = 100$. Figure 3 (first row) reports the relative approximation error, defined as the absolute difference between the estimated and true distances divided by the true distance, averaged over 10 independent runs. We compare Stream-SW with SW using random sampling to retain $3k + 2\log(n/(2k/3))$ samples. The results show that Stream-SW achieves substantially lower relative error while requiring fewer retained samples.

Next, we repeat the same experiment using two mixtures of Gaussians of 3 components. We again investigate the approximation error and the number of retained samples as $k$ and $n$ vary. Figure 3 (second row) shows the relative approximation error over 10 independent runs. As before, we compare Stream-SW with SW using random sampling to retain $3k + 2\log(n/(2k/3))$ samples, corresponding to the upper bound on the KLL sketch size. The results exhibit the same trend: Stream-SW consistently achieves lower relative error while retaining fewer samples. Similar behavior is also observed for empirical distributions over MNIST digits (LeCun et al., 1998), as reported in Appendix C.

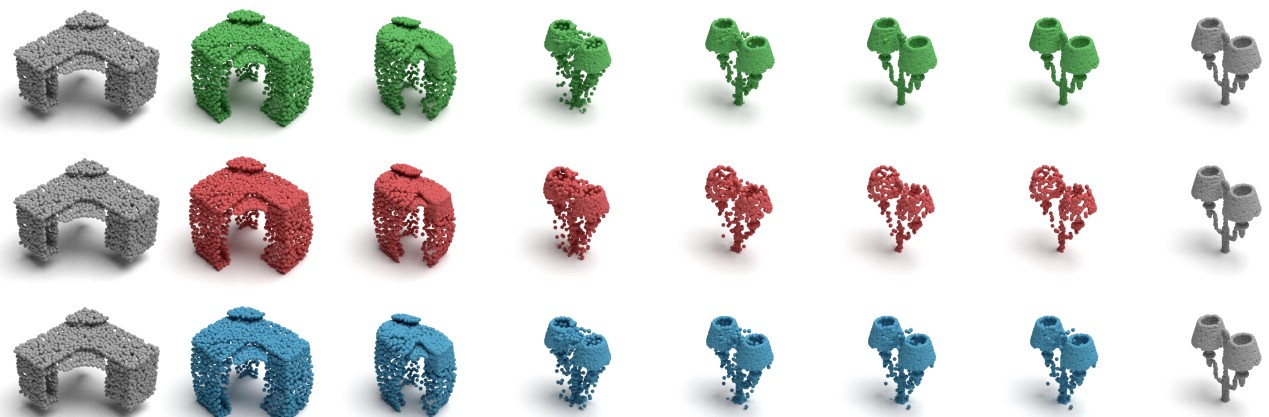

*Figure 4.* Gradient flows from Full SW, SW with random sampling, and Stream-SW in turn.

*Table 1.* Summary of Wasserstein-2 scores (Flamary et al., 2021) (multiplied by $10^3$) from three different runs.

| Loss | Step 0 | Step 500 | Step 1000 | Step 2000 | Step 3000 | Step 4000 | Step 5000 |
|------|--------|----------|-----------|-----------|-----------|-----------|-----------|
| SW L=100 (Full) | $204.83 \pm 0.0$ | $89.12 \pm 0.0$ | $26.78 \pm 0.0$ | $1.9 \pm 0.0$ | $0.94 \pm 0.0$ | $0.87 \pm 0.0$ | $0.84 \pm 0.0$ |
| SW L=1000 (Full) | $204.83 \pm 0.0$ | $88.98 \pm 0.0$ | $26.62 \pm 0.0$ | $1.63 \pm 0.0$ | $0.31 \pm 0.0$ | $0.08 \pm 0.0$ | $0.01 \pm 0.0$ |
| SW L=100 k=100 (Sampling) | $204.83 \pm 0.0$ | $90.83 \pm 1.3$ | $31.1 \pm 1.05$ | $9.8 \pm 1.33$ | $9.37 \pm 1.42$ | $9.37 \pm 1.45$ | $9.36 \pm 1.46$ |
| Stream-SW L=100 k=100 | $204.83 \pm 0.0$ | $\mathbf{89.14 \pm 0.04}$ | $\mathbf{26.94 \pm 0.07}$ | $\mathbf{3.01 \pm 0.03}$ | $\mathbf{2.1 \pm 0.02}$ | $\mathbf{1.97 \pm 0.03}$ | $\mathbf{1.93 \pm 0.03}$ |
| SW L=100 k=200 (Sampling) | $204.83 \pm 0.0$ | $90.32 \pm 0.92$ | $29.54 \pm 0.62$ | $7.19 \pm 2.53$ | $6.68 \pm 2.84$ | $6.64 \pm 2.89$ | $6.63 \pm 2.9$ |
| Stream-SW L=100 k=200 | $204.83 \pm 0.0$ | $\mathbf{89.14 \pm 0.01}$ | $\mathbf{26.84 \pm 0.01}$ | $\mathbf{2.31 \pm 0.01}$ | $\mathbf{1.46 \pm 0.01}$ | $\mathbf{1.38 \pm 0.01}$ | $\mathbf{1.36 \pm 0.02}$ |
| SW L=1000 k=100 (Sampling) | $204.83 \pm 0.0$ | $90.69 \pm 1.28$ | $30.9 \pm 1.08$ | $9.58 \pm 1.41$ | $9.42 \pm 1.57$ | $9.53 \pm 1.58$ | $9.59 \pm 1.58$ |
| Stream-SW L=1000 k=100 | $204.83 \pm 0.0$ | $\mathbf{89.08 \pm 0.01}$ | $\mathbf{26.85 \pm 0.01}$ | $\mathbf{2.5 \pm 0.01}$ | $\mathbf{1.17 \pm 0.01}$ | $\mathbf{0.93 \pm 0.03}$ | $\mathbf{0.85 \pm 0.03}$ |
| SW L=1000 k=200 (Sampling) | $204.83 \pm 0.0$ | $90.17 \pm 0.92$ | $29.37 \pm 0.63$ | $6.99 \pm 2.61$ | $6.57 \pm 2.98$ | $6.59 \pm 3.05$ | $6.59 \pm 3.06$ |
| Stream-SW L=1000 k=200 | $204.83 \pm 0.0$ | $\mathbf{89.0 \pm 0.0}$ | $\mathbf{26.67 \pm 0.01}$ | $\mathbf{1.93 \pm 0.01}$ | $\mathbf{0.73 \pm 0.01}$ | $\mathbf{0.54 \pm 0.01}$ | $\mathbf{0.48 \pm 0.01}$ |

*Table 2.* Test accuracy via K-nearest neighbors.

| $L$ | SW (Full) | SW (Random Sampling) | | | | Stream-SW | | | |
|-----|-----------|--------|--------|--------|--------|--------|--------|--------|--------|
| | | $k=5$ | $k=10$ | $k=20$ | $k=50$ | $k=5$ | $k=10$ | $k=20$ | $k=50$ |
| $L=5$ | 68.33 | 39.33 | 51.67 | 56 | 61.33 | 45.53 | 45.33 | 63.33 | 68.67 |
| $L=10$ | 73.33 | 47.67 | 55.33 | 62 | 68 | 52.67 | 69.33 | 73 | 73 |
| $L=50$ | 77.33 | 54 | 63.67 | 67.67 | 73.33 | 70 | 73.67 | 76.33 | 77.67 |
| $L=100$ | 77.67 | 56.67 | 62 | 68 | 72.67 | 72.33 | 76.67 | 77.67 | 77.67 |

*Table 3.* Delay (Early) time for detecting the transition in MSRC-12 that corresponds to four users.

| Guest 1 | | Guest 2 | | Guest 3 | | Guest 4 | |
|---------|-----------|---------|-----------|---------|-----------|---------|-----------|
| SW | Stream-SW | SW | Stream-SW | SW | Stream-SW | SW | Stream-SW |
| 100 | 50 | 100 | 10 | 49 | 24 | 100 | 32 |

### 4.2. Streaming Point-cloud Classification

We follow the setup in (Li et al., 2024). In particular, we select (stratified sampling from 10 categories) 500 point clouds as the training set and 300 point clouds as the testing set from the ModelNet10 dataset (Wu et al., 2015). For each point cloud, we sample 500 points. After that, we use the K-NN algorithm with $K=5$ neighbors to evaluate the classification accuracy on the testing set. We vary the sketch size $k \in \{5, 10, 20, 50\}$ and the number of projections $L \in \{5, 10, 50, 100\}$.

We show the result for Stream-SW, SW with the random sampling of $3k + 2\log(n/(2k/3))$ samples, and Full SW

(keeping all samples) in Table 2. From the table, we see that Stream-SW can lead to considerably higher classification accuracy than SW for any choices of $L$ and $k$, especially when having small $k$ e.g., 5, 10. Stream-SW can lead to comparable accuracies when having large enough $k$ e.g., 20. When the number of projections is large enough e.g., $L=100$, Stream-SW can lead to comparable accuracy to Full SW with $k=10$.

### 4.3. Streaming Gradient Flow

We model a distribution $\mu(t)$ flowing with time $t$ along the gradient flow of a loss functional $\mu(t) \rightarrow SW_2(\mu(t), \nu)$ that drives it towards a target distribution $\nu$ (Santambrogio, 2015). We consider discrete flows, namely, $\mu(t) = \frac{1}{n}\sum_{i=1}^{n}\delta_{x_i(t)}$ and $\nu = \frac{1}{n}\sum_{i=1}^{n}\delta_{y_i}$. Here, $\nu$ is assumed to be received from a sample stream. We refer to Appendix B for the detail about one-sided streaming sliced Wasserstein, which is the case here. We choose $\mu(0)$ and $\nu$ as two point-cloud shapes in the ShapeNet Core-55 dataset (Chang et al., 2015). After that, we solve the flows by using the Euler scheme with 5000 iterations, step size 0.001, and approximated gradient as discussed in Appendix B. We show the Wasserstein-2 scores (Flamary et al., 2021) in 3 different runs for Stream-SW, SW with random sam-

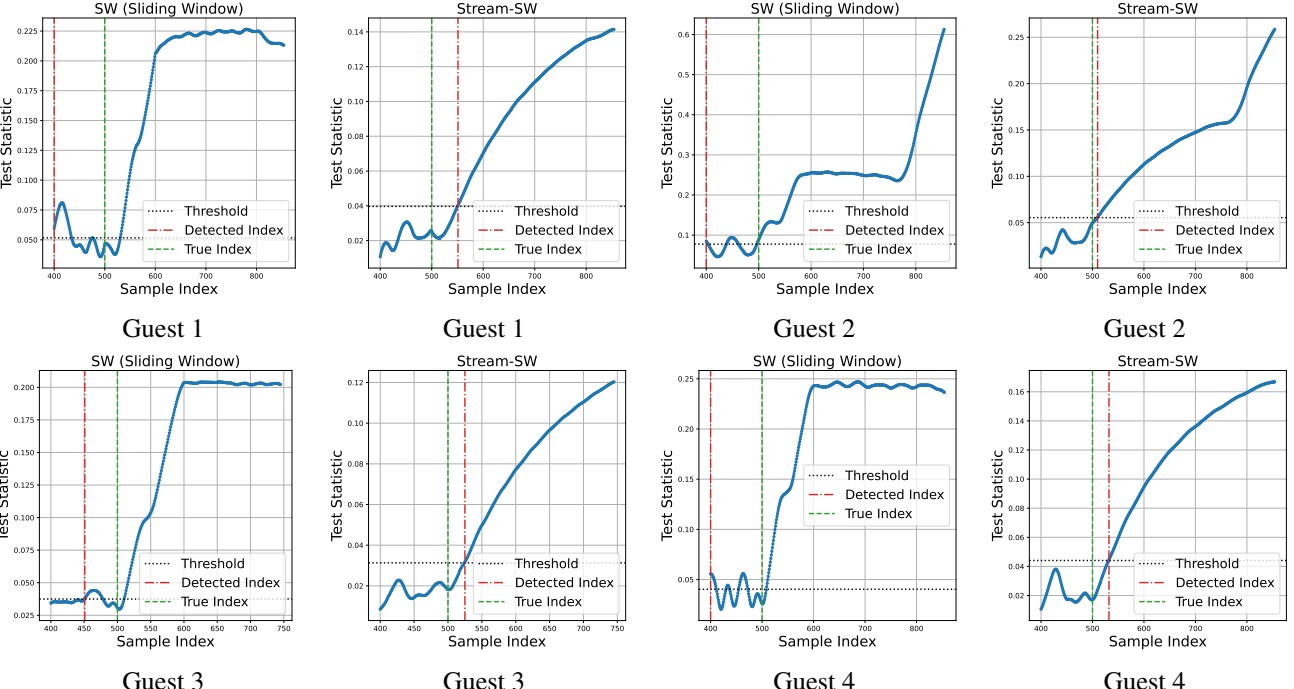

*Figure 5.* The figure shows the decision statistics, decision thresholds (null statistics), the true time index (where the change in action happens), and the detected indices, for both two approaches i.e., SW (Sliding window) and Stream-SW for the four guests.

pling of $3k + 2\log(n/(2k/3))$ samples, and Full SW with $L \in \{100, 1000\}$ and $k \in \{100, 200\}$ in Table 1. We see that Stream-SW makes flows converge faster than SW with random sampling. We strengthen the claim by qualitative comparison in Figure 4 with $L = 1000$ and $k = 200$. We visualize flows in other settings in Appendix C.

### 4.4. Streaming Change Point Detection

We follow the same setup as in (Wang et al., 2022) using the Microsoft Research Cambridge-12 (MSRC-12) Kinect gesture dataset (Fothergill et al., 2012). After preprocessing, the dataset contains actions from four subjects, each with 855 samples in $\mathbb{R}^{60}$, where a change from bending to throwing occurs at time index 500. Traditional change-point detection relies on a sliding window, but it is sensitive to the choice of window size, may miss or dilute anomalies near window boundaries, incurs computational overhead for small steps, and often ignores long-range temporal dependencies. In contrast, we compare the distribution before detection with the full observed distribution over time, declaring a change when a significant discrepancy appears. With Stream-SW, this can be done directly in a streaming manner.

Concretely, the sliding-window baseline uses window size $W = 100$. A null distribution is built by computing 1000 distances between random windows from pre-change data (before time 300), yielding a threshold that controls the false alarm rate at $\alpha = 0.05$. For Stream-SW, we similarly compute the SW distance between two random subsets

of size 100 via bootstrap on pre-change data, and trigger detection when it exceeds the threshold. Table 3 reports detection delays. Overall, Stream-SW achieves substantially better detection performance than the sliding-window baseline. Figure 5 visualizes the decision statistics, thresholds, true change point, and detected points across all subjects, confirming the advantage of the proposed approach. We note that performance can be further improved using kernel enhancements, optimized projections, and advanced distributional operators (Wang et al., 2022; Deshpande et al., 2019; Kolouri et al., 2019).

## 5. Conclusion

We have presented streaming sliced Wasserstein (Stream-SW) which is the first streaming estimator for sliced Wasserstein distance from sample streams. The key idea of Stream-SW is to utilize streaming quantile function approximation based on quantile sketches. We have discussed approximation guarantees and computational complexities of Stream-SW. On the experimental side, we investigate the approximation error of Stream-SW, and compare it with SW using random sampling in streaming point-cloud classification, streaming gradient flow, and streaming change point detection. Future works will extend Stream-SW to other variants of SW such as generalized sliced Wasserstein (Kolouri et al., 2019), spherical sliced Wasserstein (Bonet et al., 2023; Tran et al., 2024), and sliced Wasserstein on manifolds (Bonet et al., 2025; Nguyen & Mueller, 2026).

## Impact Statement

This paper presents work to advance the field of Machine Learning. There are potential societal consequences of our work, none of which we feel must be highlighted.

## Acknowledgements

We would like to thank Professor Joydeep Ghosh for his suggestion on the change point detection application, Dr. Tuan Pham for his insightful discussion during the course of this project, and ICML reviewers for constructive feedback during the peer-review process.

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

# Supplement to "Streaming Sliced Optimal Transport"

We first provide skipped technical proofs in Appendix A. We then provide additional materials in Appendix B. Additional experimental results in streaming Gaussian comparison, and streaming gradient flows in Appendix C. We report the detail of computational device for experiments in Appendix D.

## A. Proofs

### A.1. Proof of Proposition 1

We denote $\mathcal{S}$ as the quantile sketch with initial size $k > 1$ constructed from streaming samples $x_1, \ldots, x_n$ from $\mu \in \mathcal{P}(\mathbb{R})$. We recall that when the top compactor was created, the second compactor from the top compacted its elements at least once. Therefore $n \geq k_{H-1} w_{H-1} = k_{H-1} 2^{H-2}$ which gives $H \leq \log(n/k_{H-1}) + 2 \leq \log((3n)/(2k)) + 2$. Since the sketch carries total weight exactly $n$ and each compactor holds at most $k_h$ items, $n \leq \sum_{h=1}^{H} k_h w_h \leq k \sum_{h=1}^{H} (2/3)^{H-h} 2^{h-1} + \sum_{h=1}^{H} 2^{h-1} \leq 2^H \left( \frac{3k}{4} + 1 \right)$, which gives the complementary lower bound $2^H \geq n/(\frac{3k}{4} + 1)$. Using this together with $k_h \geq k(2/3)^{H-h}$ and $w_h = 2^{h-1}$, we can bound the number of compact operations at height $h$ by

$$m_h \leq \frac{n}{k_h w_h} \leq \frac{2n}{k 2^H} 3^{H-h} \leq \left( \frac{3}{2} + \frac{2}{k} \right) 3^{H-h} \leq 6 \cdot 3^{H-h-1} \tag{20}$$

for $k \geq 4$.

Following Karnin et al. (2016), let $R(x, h)$ denote the rank of $x$ among the items yielded by the compactor at height $h$ together with all items stored in compactors of height $h' \leq h$ at the end of the stream, and set $F_{\mathcal{S},h}(x) := R(x, h)/n$, so that $F_{\mathcal{S},0} = F_n$ is the exact empirical CDF and $F_{\mathcal{S},H} = F_{\mathcal{S}}$. Define

$$error(x, h) = F_{\mathcal{S},h}(x) - F_{\mathcal{S},h-1}(x), \tag{21}$$

the total change in the CDF value at $x$ due to level $h$. Each compaction operation in level $h$ either leaves the value of the CDF of $x$ unchanged or adds $w_h/n$ or subtracts $w_h/n$ with equal probability. Therefore $error(x, h) = \sum_{i=1}^{m_h} \frac{w_h}{n} \xi_{i,h}$ where $\xi_{i,h}$ are independent random variables with $\mathbb{E}[\xi_{i,h}] = 0$ and $|\xi_{i,h}| \leq 1$. The total error telescopes:

$$F_{\mathcal{S}}(x) - F_n(x) = \sum_{h=1}^{H} error(x, h) = \sum_{h=1}^{H} \sum_{i=1}^{m_h} \frac{w_h}{n} \xi_{i,h}. \tag{22}$$

By Hoeffding's inequality, for every fixed $x \in \mathbb{R}$,

$$\mathbb{P} \left( |F_n(x) - F_{\mathcal{S}}(x)| > \epsilon \right) \leq 2 \exp \left( - \frac{n^2 \epsilon^2}{2 \sum_{h=1}^{H} \sum_{i=1}^{m_h} w_h^2} \right). \tag{23}$$

Using $w_h^2 = 2^{2h-2}$ (weight $w_h = 2^{h-1}$) and $m_h \leq 6 \cdot 3^{H-h-1}$,

$$\sum_{h=1}^{H} \sum_{i=1}^{m_h} w_h^2 \leq 6 \sum_{h=1}^{H} 3^{H-h-1} 2^{2h-2} = \frac{6 \cdot 3^{H-1}}{4} \sum_{h=1}^{H} \left( \frac{4}{3} \right)^h \leq 2 \cdot 2^{2H} \leq 2 \cdot 2^{2 \log((3n)/(2k)) + 4} = \frac{72 n^2}{k^2}, \tag{24}$$

using $H \leq \log((3n)/(2k)) + 2$ in the last step. Overall, for every fixed $x$,

$$\mathbb{P} \left( |F_n(x) - F_{\mathcal{S}}(x)| > \epsilon \right) \leq 2 \exp \left( -C k^2 \epsilon^2 \right), \qquad C = \frac{1}{144}. \tag{25}$$

**Uniformity over $x$.** Condition on $x_1, \ldots, x_n$, so that $F_n$ is deterministic; the compaction coins remain independent of the data. For $\epsilon \in (0, 1]$ let $N = \lceil 2/\epsilon \rceil$ and define grid points $z_j := F_n^{-1}(j/N)$ for $j = 1, \ldots, N$, together with $z_0 := \inf\{x : F_n(x) > 0\}$. Write $z_j^-$ for a point immediately to the left of $z_j$; since $F_n$ and $F_{\mathcal{S}}$ are step functions with jumps only at stream points, $F_n(z_j^-)$ and $F_{\mathcal{S}}(z_j^-)$ are attained at fixed points. By definition of the generalized inverse, $F_n(z_j) \geq j/N$ and $F_n(z_j^-) \leq j/N$.

If $x \geq z_N$ then $F_n(x) = F_{\mathcal{S}}(x) = 1$, and if $x < z_0$ then $F_n(x) = F_{\mathcal{S}}(x) = 0$, since the sketch stores a subset of the stream with total weight $n$. Otherwise $x \in [z_j, z_{j+1})$ for some $j \in \{0, \dots, N-1\}$, and monotonicity of both functions gives

$$F_{\mathcal{S}}(x) - F_n(x) \leq \left[F_{\mathcal{S}}(z_{j+1}^-) - F_n(z_{j+1}^-)\right] + \left[F_n(z_{j+1}^-) - F_n(z_j)\right], \tag{26}$$

$$F_{\mathcal{S}}(x) - F_n(x) \geq \left[F_{\mathcal{S}}(z_j) - F_n(z_j)\right] - \left[F_n(z_{j+1}^-) - F_n(z_j)\right]. \tag{27}$$

Since $F_n(z_{j+1}^-) \leq (j+1)/N$ and $F_n(z_j) \geq j/N$, the bracketed increment is at most $1/N \leq \epsilon/2$, and therefore

$$\sup_{x \in \mathbb{R}} \left|F_n(x) - F_{\mathcal{S}}(x)\right| \leq \max_{j=0,\dots,N} \left\{\left|F_n(z_j) - F_{\mathcal{S}}(z_j)\right|, \left|F_n(z_j^-) - F_{\mathcal{S}}(z_j^-)\right|\right\} + \frac{\epsilon}{2}. \tag{28}$$

Applying the pointwise bound at each of the $2(N+1) \leq 8/\epsilon$ evaluation points (using $\epsilon \leq 1$) with target accuracy $\epsilon/2$, and taking a union bound,

$$\mathbb{P}\left(\sup_{x \in \mathbb{R}} \left|F_n(x) - F_{\mathcal{S}}(x)\right| > \epsilon\right) \leq \frac{8}{\epsilon} \cdot 2 \exp\left(-Ck^2(\epsilon/2)^2\right) \leq \frac{16}{\epsilon} \exp\left(-\frac{Ck^2\epsilon^2}{4}\right). \tag{29}$$

**Compact support.** When $\mu$ has compact support of diameter at most $R$, the integrand vanishes outside that interval, so $\int_{\mathbb{R}} |F_n - F_{\mathcal{S}}| \mathrm{d}x \leq R \sup_x |F_n - F_{\mathcal{S}}|$, and applying the display above with $\epsilon/R$ in place of $\epsilon$,

$$\mathbb{P}\left(\int_{\mathbb{R}} |F_n(x) - F_{\mathcal{S}}(x)| \mathrm{d}x > \epsilon\right) \leq \frac{16R}{\epsilon} \exp\left(-\frac{Ck^2\epsilon^2}{4R^2}\right) \leq \frac{A_R}{\epsilon} \exp(-C_R k^2 \epsilon^2), \tag{30}$$

for constants $A_R, C_R > 0$ depending only on $R$, which completes the proof.

### A.2. Proof of Proposition 2

We denote $\mathcal{S}$ as the quantile sketch with initial size $k > 1$ constructed from streaming samples $x_1, \dots, x_n$ from $\mu \in \mathcal{P}(\mathbb{R})$. Let define $F_n(y) = \frac{1}{n} \sum_{i=1}^n I(x_i \leq y)$, $F_n^{-1}(q) = \inf\{x : F_n(x) \geq q\}$, and $F_{\mathcal{S}_\mu}^{-1}(q) = \inf\{x : F_{\mathcal{S}_\mu}(x) \geq q\}$. For any $q \in [0, 1]$, we have:

$$\mathbb{P}\left(F_{\mathcal{S}}^{-1}(q) > F_n^{-1}(q) + \epsilon\right) = \mathbb{P}\left(F_{\mathcal{S}}(F_n^{-1}(q) + \epsilon) < q\right)$$
$$= \mathbb{P}\left(F_{\mathcal{S}}(F_n^{-1}(q) + \epsilon) - F_n(F_n^{-1}(q) + \epsilon) < q - F_n(F_n^{-1}(q) + \epsilon)\right) \tag{31}$$

From Appendix A.1, we know that $F_{\mathcal{S}}(F_n^{-1}(q) + \epsilon) - F_n(F_n^{-1}(q) + \epsilon)$ is the sum of Bernoulli random variables, by the Hoeffding's inequality, we have:

$$\mathbb{P}\left(F_{\mathcal{S}}^{-1}(q) > F_n^{-1}(q) + \epsilon\right) \leq \exp\left(-Ck^2(F_n(F_n^{-1}(q) + \epsilon) - q)^2\right), \tag{32}$$

for a constant $C > 0$. Similarly, we have:

$$\mathbb{P}\left(F_{\mathcal{S}}^{-1}(q) \leq F_n^{-1}(q) - \epsilon\right) = \mathbb{P}\left(F_{\mathcal{S}}(F_n^{-1}(q) - \epsilon) \geq q\right)$$
$$= \mathbb{P}\left(F_{\mathcal{S}}(F_n^{-1}(q) - \epsilon) - F_n(F_n^{-1}(q) - \epsilon) \geq q - F_n(F_n^{-1}(q) - \epsilon)\right)$$
$$\leq \exp\left(-Ck^2(q - F_n(F_n^{-1}(q) - \epsilon))^2\right). \tag{33}$$

By the union bound, we have:

$$\mathbb{P}\left(\left|F_{\mathcal{S}}^{-1}(q) - F_n^{-1}(q)\right| > \epsilon\right) \leq \mathbb{P}\left(F_{\mathcal{S}}^{-1}(q) - F_n^{-1}(q) > \epsilon\right) + \mathbb{P}\left(F_{\mathcal{S}}^{-1}(q) - F_n^{-1}(q) \leq -\epsilon\right)$$
$$\leq \exp\left(-Ck^2(F_n(F_n^{-1}(q) + \epsilon) - q)^2\right) + \exp\left(-Ck^2(q - F_n(F_n^{-1}(q) - \epsilon))^2\right)$$
$$\leq 2 \exp\left(-Ck^2\gamma_{q,\epsilon}^2\right), \tag{34}$$

where $\gamma_{q,\epsilon} = \min\left\{q - F_n(F_n^{-1}(q) - \epsilon), F_n(F_n^{-1}(q) + \epsilon) - q\right\}$.

**Compact support.** When having compact support (at most $R$), we have:

$$\mathbb{P}\left(\int_0^1 \left|F_{\mathcal{S}}^{-1}(q) - F_n^{-1}(q)\right| \mathrm{d}q > \epsilon\right) = \mathbb{P}\left(\int_{\mathbb{R}} |F_{\mathcal{S}}(x) - F_n(x)| \mathrm{d}x > \epsilon\right) \leq \frac{A_R}{\epsilon} \exp\left(-C_R k^2 \epsilon^2\right), \tag{35}$$

where the last inequality is proven in Appendix A.1, and the identity is the standard duality between $L^1$-distance of quantile functions and of CDFs on $[0, 1] \times \mathbb{R}$.

### A.3. Proof of Proposition 3

In one dimension, the $p$-Wasserstein distance admits the quantile representation

$$W_p^p(\mu_n, \nu_m) = \int_0^1 \left| F_{\mu_n}^{-1}(q) - F_{\nu_m}^{-1}(q) \right|^p \mathrm{d}q, \tag{36}$$

and similarly for $\widetilde{W}_p^p(\mu_n, \nu_m; \mathcal{S}_{\mu_n, k_1}, \mathcal{S}_{\nu_m, k_2})$ with $F_{\mathcal{S}_{\mu_n, k_1}}^{-1}, F_{\mathcal{S}_{\nu_m, k_2}}^{-1}$ in place of $F_{\mu_n}^{-1}, F_{\nu_m}^{-1}$.

**Empirical approximation.** For $a = |F_{\mathcal{S}_{\mu_n, k_1}}^{-1}(q) - F_{\mathcal{S}_{\nu_m, k_2}}^{-1}(q)|$ and $b = |F_{\mu_n}^{-1}(q) - F_{\nu_m}^{-1}(q)|$, both lying in $[0, R]$, the mean value theorem gives the two-sided bound

$$|a^p - b^p| \leq p \max\{a, b\}^{p-1}|a - b| \leq pR^{p-1}|a - b|. \tag{37}$$

(Absolute values are needed on both sides here: the one-sided inequality $a^p - b^p \leq p \max\{a, b\}^{p-1}(a - b)$ fails when $a < b$.) By the reverse triangle inequality,

$$|a - b| \leq \left| F_{\mathcal{S}_{\mu_n, k_1}}^{-1}(q) - F_{\mu_n}^{-1}(q) \right| + \left| F_{\mathcal{S}_{\nu_m, k_2}}^{-1}(q) - F_{\nu_m}^{-1}(q) \right|. \tag{38}$$

Integrating over $q \in [0, 1]$,

$$\left| \widetilde{W}_p^p(\mu_n, \nu_m; \mathcal{S}_{\mu_n, k_1}, \mathcal{S}_{\nu_m, k_2}) - W_p^p(\mu_n, \nu_m) \right| \leq pR^{p-1} \Big( \underbrace{\int_0^1 |F_{\mathcal{S}_{\mu_n, k_1}}^{-1}(q) - F_{\mu_n}^{-1}(q)| \mathrm{d}q}_{=: I_1} + \underbrace{\int_0^1 |F_{\mathcal{S}_{\nu_m, k_2}}^{-1}(q) - F_{\nu_m}^{-1}(q)| \mathrm{d}q}_{=: I_2} \Big). \tag{39}$$

Let $\delta = \epsilon/(2pR^{p-1})$. If the left-hand side exceeds $\epsilon$, then $I_1 > \delta$ or $I_2 > \delta$. By Proposition 2 (compact-support case) and a union bound,

$$\mathbb{P}\Big( \big| \widetilde{W}_p^p - W_p^p(\mu_n, \nu_m) \big| > \epsilon \Big) \leq \frac{2A_R}{\delta} \exp\big( - C_R \min\{k_1, k_2\}^2 \delta^2 \big) \leq \frac{A_{p,R}}{\epsilon} \exp\big( - C_{p,R} \min\{k_1^2, k_2^2\}\epsilon^2 \big), \tag{40}$$

with $C_{p,R} = C_R/(4p^2 R^{2(p-1)})$ and $A_{p,R} = 4pR^{p-1}A_R$.

**Population approximation.** We decompose, by the triangle inequality,

$$\left| \widetilde{W}_p^p(\mu_n, \nu_m; \mathcal{S}_{\mu_n, k_1}, \mathcal{S}_{\nu_m, k_2}) - W_p^p(\mu, \nu) \right| \leq A + B, \tag{41}$$

where $A := \big| \widetilde{W}_p^p - W_p^p(\mu_n, \nu_m) \big|$ and $B := \big| W_p^p(\mu_n, \nu_m) - W_p^p(\mu, \nu) \big|$, so that $\mathbb{P}(A + B > \epsilon) \leq \mathbb{P}(A > \epsilon/2) + \mathbb{P}(B > \epsilon/2)$.

The first term is bounded by the empirical approximation above with $\epsilon/2$ in place of $\epsilon$. For the second, the identical mean value theorem and triangle inequality argument used above, applied directly to $(\mu_n, \nu_m)$ versus $(\mu, \nu)$ (both pairs supported on the diameter-$R$ interval), gives, with $C'_{p,R} := pR^{p-1}$,

$$B \leq C'_{p,R}\big( W_1(\mu_n, \mu) + W_1(\nu_m, \nu) \big), \tag{42}$$

using the exact one-dimensional identity $\int_0^1 |F_\alpha^{-1} - F_\beta^{-1}| \mathrm{d}q = W_1(\alpha, \beta)$. Hence, by a union bound,

$$\mathbb{P}\big(B > \tfrac{\epsilon}{2}\big) \leq \mathbb{P}\Big( W_1(\mu_n, \mu) > \tfrac{\epsilon}{4C'_{p,R}} \Big) + \mathbb{P}\Big( W_1(\nu_m, \nu) > \tfrac{\epsilon}{4C'_{p,R}} \Big). \tag{43}$$

Since $W_1(\mu_n, \mu) = \int_{\mathbb{R}} |F_{\mu_n} - F_\mu| \mathrm{d}x \leq R \sup_x |F_{\mu_n} - F_\mu|$, the Dvoretzky–Kiefer–Wolfowitz inequality gives $\mathbb{P}(W_1(\mu_n, \mu) > \delta) \leq 2 \exp(-2n\delta^2/R^2)$. Setting $\delta = \epsilon/(4C'_{p,R})$,

$$\mathbb{P}\big(B > \tfrac{\epsilon}{2}\big) \leq 4 \exp\left( -\frac{\min\{n, m\}\epsilon^2}{8R^2 C'^2_{p,R}} \right). \tag{44}$$

Combining the two bounds and using $e^{-a} + e^{-b} \leq 2e^{-\min\{a,b\}}$,

$$\mathbb{P}\Big( \big| \widetilde{W}_p^p - W_p^p(\mu, \nu) \big| > \epsilon \Big) \leq \Big( \frac{a_{p,R}}{\epsilon} + 8 \Big) \exp\big( -c_{p,R}\epsilon^2 \min\{k_1^2, k_2^2, n, m\} \big), \tag{45}$$

for constants $a_{p,R}, c_{p,R} > 0$ depending only on $p$ and $R$, which completes the proof.

### A.4. Proof of Proposition 4

For any $x \in \mathbb{R}$, by the triangle inequality,

$$
\begin{aligned}
\left| \widetilde{T}(x) - T_{\mu \to \nu}(x) \right| &= \left| F_{\mathcal{S}_{\nu_m, k_2}}^{-1} (F_{\mathcal{S}_{\mu_n, k_1}}(x)) - F_\nu^{-1}(F_\mu(x)) \right| \\
&\le \underbrace{\left| F_\nu^{-1}(F_{\mathcal{S}_{\mu_n, k_1}}(x)) - F_\nu^{-1}(F_\mu(x)) \right|}_{=:A(x)} + \underbrace{\left| F_{\mathcal{S}_{\nu_m, k_2}}^{-1}(F_{\mathcal{S}_{\mu_n, k_1}}(x)) - F_\nu^{-1}(F_{\mathcal{S}_{\mu_n, k_1}}(x)) \right|}_{=:B(x)}.
\end{aligned}
\tag{46}
$$

Since $f_\nu \ge a > 0$, $F_\nu^{-1}$ is $1/a$-Lipschitz on $[0,1]$, so $A(x) \le \frac{1}{a} |F_{\mathcal{S}_{\mu_n, k_1}}(x) - F_\mu(x)|$, and integrating against $\mu$,

$$
\int_{\mathbb{R}} A(x) \, d\mu(x) \le \frac{1}{a} \sup_{x \in \mathbb{R}} \left| F_{\mathcal{S}_{\mu_n, k_1}}(x) - F_\mu(x) \right| \le \frac{1}{a} \sup_x \left| F_{\mathcal{S}_{\mu_n, k_1}}(x) - F_{\mu_n}(x) \right| + \frac{1}{a} \sup_x \left| F_{\mu_n}(x) - F_\mu(x) \right|.
\tag{47}
$$

By the uniform CDF bound established in the proof of Proposition 1 and the Dvoretzky–Kiefer–Wolfowitz inequality,

$$
\mathbb{P}\left( \int_{\mathbb{R}} A(x) \, d\mu(x) > \frac{\epsilon}{2} \right) \le \left( \frac{A_a}{\epsilon} + 4 \right) \exp\left( -C_a \, \epsilon^2 \min\{k_1^2, n\} \right),
\tag{48}
$$

for constants $A_a, C_a > 0$ depending only on $a$.

For $B(x)$, let $\eta := \sup_x |F_{\mathcal{S}_{\nu_m, k_2}}(x) - F_\nu(x)|$ and fix $q \in [0,1]$, $y := F_{\mathcal{S}_{\nu_m, k_2}}^{-1}(q)$, so $F_{\mathcal{S}_{\nu_m, k_2}}(y^-) \le q \le F_{\mathcal{S}_{\nu_m, k_2}}(y)$. Since $\nu$ is absolutely continuous, $F_\nu$ is continuous and $F_\nu(y) = F_\nu(y^-)$, so

$$
F_\nu(y) \ge F_{\mathcal{S}_{\nu_m, k_2}}(y) - \eta \ge q - \eta, \qquad F_\nu(y) \le F_{\mathcal{S}_{\nu_m, k_2}}(y^-) + \eta \le q + \eta,
\tag{49}
$$

i.e. $|F_\nu(y) - q| \le \eta$. As $F_\nu^{-1}$ is $1/a$-Lipschitz, $|y - F_\nu^{-1}(q)| \le \eta/a$, hence

$$
\sup_{q \in [0,1]} \left| F_{\mathcal{S}_{\nu_m, k_2}}^{-1}(q) - F_\nu^{-1}(q) \right| \le \frac{1}{a} \sup_x \left| F_{\mathcal{S}_{\nu_m, k_2}}(x) - F_\nu(x) \right| \le \frac{1}{a} \sup_x \left| F_{\mathcal{S}_{\nu_m, k_2}}(x) - F_{\nu_m}(x) \right| + \frac{1}{a} \sup_x \left| F_{\nu_m}(x) - F_\nu(x) \right|.
\tag{50}
$$

Since $F_{\mathcal{S}_{\mu_n, k_1}}(x) \in [0,1]$ for all $x$ and $\mu(\mathbb{R}) = 1$, $\int_{\mathbb{R}} B(x) \, d\mu(x) \le \sup_q |F_{\mathcal{S}_{\nu_m, k_2}}^{-1}(q) - F_\nu^{-1}(q)|$, so again by the uniform CDF bound and DKW,

$$
\mathbb{P}\left( \int_{\mathbb{R}} B(x) \, d\mu(x) > \frac{\epsilon}{2} \right) \le \left( \frac{A_a}{\epsilon} + 4 \right) \exp\left( -C_a \, \epsilon^2 \min\{k_2^2, m\} \right).
\tag{51}
$$

By the union bound,

$$
\mathbb{P}\left( \int_{\mathbb{R}} \left| \widetilde{T}(x) - T_{\mu \to \nu}(x) \right| d\mu(x) > \epsilon \right) \le \left( \frac{a_{a,R}}{\epsilon} + 8 \right) \exp\left( -c_a \, \epsilon^2 \min\{k_1^2, k_2^2, n, m\} \right),
\tag{52}
$$

for constants $a_{a,R}, c_a > 0$ depending only on $a$ and $R$.

### A.5. Proof of Corollary 1

By definition of the sliced Wasserstein distance,

$$
SW_p^p(\mu_n, \nu_m) = \mathbb{E}_{\theta \sim \sigma}\left[ W_p^p(\theta \sharp \mu_n, \theta \sharp \nu_m) \right],
\tag{53}
$$

$$
\widetilde{SW}_p^p(\mu_n, \nu_m; k_1, k_2) = \mathbb{E}_{\theta \sim \sigma}\left[ \widetilde{W}_p^p(\theta \sharp \mu_n, \theta \sharp \nu_m; S_{\theta \sharp \mu_n, k_1}, S_{\theta \sharp \nu_m, k_2}) \right].
\tag{54}
$$

By the triangle and Jensen inequalities,

$$
\left| \widetilde{SW}_p^p(\mu_n, \nu_m; k_1, k_2) - SW_p^p(\mu_n, \nu_m) \right| \le \mathbb{E}_\theta \left[ X_\theta \right], \qquad X_\theta := \left| \widetilde{W}_p^p(\theta \sharp \mu_n, \theta \sharp \nu_m) - W_p^p(\theta \sharp \mu_n, \theta \sharp \nu_m) \right|.
\tag{55}
$$

Fix $\theta$. Since $x \mapsto \theta^\top x$ is 1-Lipschitz, $\theta \sharp \mu_n$ and its sketch are supported on an interval of diameter at most $R$. We bound $\mathbb{E}[X_\theta]$ via the variance computation from the proof of Proposition 1. Writing $F_{\mathcal{S}} - F_n = \sum_h \sum_i \frac{w_h}{n} \xi_{i,h}$ with independent,

mean-zero, $|\xi_{i,h}| \leq 1$ terms and $\sum_h \sum_i w_h^2 \leq 72n^2/k^2$, we get $\text{Var}(F_{\mathcal{S}}(x) - F_n(x)) \leq 72/k^2$ uniformly in $x$. Using the identity $\int_0^1 |F_\alpha^{-1}(q) - F_\beta^{-1}(q)|\mathrm{d}q = \int_{\mathbb{R}} |F_\alpha(x) - F_\beta(x)|\mathrm{d}x$, Jensen's inequality $\mathbb{E}|Z| \leq \sqrt{\text{Var}(Z)}$ for the mean-zero variable $Z = F_{\mathcal{S}}(x) - F_n(x)$, and Fubini over the diameter-$R$ support,

$$\mathbb{E}\Big[\int_0^1 \big|F_{\mathcal{S}_{\theta\sharp\mu_n,k_1}}^{-1}(q) - F_{\theta\sharp\mu_n}^{-1}(q)\big|\mathrm{d}q\Big] = \int_{\mathbb{R}} \mathbb{E}\big|F_{\mathcal{S}_{\theta\sharp\mu_n,k_1}}(x) - F_{\theta\sharp\mu_n}(x)\big|\mathrm{d}x \leq \frac{6\sqrt{2}\,R}{k_1}, \tag{56}$$

and likewise with $k_2$ for $\theta\sharp\nu_m$. Combined with the empirical-approximation bound $|\widetilde{W}_p^p - W_p^p(\theta\sharp\mu_n, \theta\sharp\nu_m)| \leq pR^{p-1}(I_1 + I_2)$ established in the proof of Proposition 3,

$$\mathbb{E}[X_\theta] \leq pR^{p-1} \cdot 6\sqrt{2}\,R\left(\frac{1}{k_1} + \frac{1}{k_2}\right) \leq C'_{p,R}\min\{k_1, k_2\}^{-1}. \tag{57}$$

Taking expectation over $\theta \sim \sigma$ gives

$$\mathbb{E}\Big[\big|\widetilde{SW}_p^p(\mu_n, \nu_m; k_1, k_2) - SW_p^p(\mu_n, \nu_m)\big|\Big] \leq C'_{p,R}\min\{k_1, k_2\}^{-1}, \tag{58}$$

which is the first claim.

**Population error.** Adding and subtracting $W_p^p(\theta\sharp\mu_n, \theta\sharp\nu_m)$ and applying the population half of Proposition 3's argument (again in expectation form) to the second difference,

$$\mathbb{E}\big[|\widetilde{W}_p^p - W_p^p(\theta\sharp\mu, \theta\sharp\nu)|\big] \leq C'_{p,R}\min\{k_1, k_2\}^{-1} + pR^{p-1}\big(\mathbb{E}[W_1(\theta\sharp\mu_n, \theta\sharp\mu)] + \mathbb{E}[W_1(\theta\sharp\nu_m, \theta\sharp\nu)]\big). \tag{59}$$

Since $\mathbb{E}[|F_n(x) - F_\mu(x)|] \leq \sqrt{F_\mu(x)(1 - F_\mu(x))/n} \leq 1/(2\sqrt{n})$ pointwise, Fubini over the diameter-$R$ support gives $\mathbb{E}[W_1(\mu_n, \mu)] \leq R/(2\sqrt{n})$, and likewise for $\nu_m$. Hence

$$\mathbb{E}\Big[\big|\widetilde{SW}_p^p(\mu_n, \nu_m; k_1, k_2) - SW_p^p(\mu, \nu)\big|\Big] \leq c'_{p,R}\min\{k_1, k_2, \sqrt{n}, \sqrt{m}\}^{-1}, \tag{60}$$

after taking expectation over $\theta \sim \sigma$, which completes the proof.

### A.6. Proof of Theorem 1

Let $\theta_1, \ldots, \theta_L$ be i.i.d. directions sampled uniformly from $\mathbb{S}^{d-1}$. Recall the Monte Carlo estimate of Stream-SW:

$$\widehat{\widetilde{SW}}_p^p(\mu_n, \nu_m; k_1, k_2, L) = \frac{1}{L}\sum_{l=1}^L \widetilde{W}_p^p\Big(\theta_l\sharp\mu_n, \theta_l\sharp\nu_m; S_{\theta_l\sharp\mu_n,k_1}, S_{\theta_l\sharp\nu_m,k_2}\Big). \tag{61}$$

By the triangle inequality:

$$\big|\widehat{\widetilde{SW}}_p^p(\mu_n, \nu_m; k_1, k_2, L) - SW_p^p(\mu, \nu)\big| \tag{62}$$
$$\leq \big|\widehat{\widetilde{SW}}_p^p(\mu_n, \nu_m; k_1, k_2, L) - \widetilde{SW}_p^p(\mu_n, \nu_m; k_1, k_2)\big| + \big|\widetilde{SW}_p^p(\mu_n, \nu_m; k_1, k_2) - SW_p^p(\mu, \nu)\big|.$$

using the Holder's inequality, we have:

$$\mathbb{E}\big|\widehat{\widetilde{SW}}_p^p(\mu_n, \nu_m; k_1, k_2, L) - \widetilde{SW}_p^p(\mu_n, \nu_m; k_1, k_2)\big|$$

$$\leq \left(\mathbb{E}\big|\widehat{\widetilde{SW}}_p^p(\mu_n, \nu_m; k_1, k_2, L) - \widetilde{SW}_p^p(\mu_n, \nu_m; k_1, k_2)\big|^2\right)^{\frac{1}{2}}$$

$$= \left(\mathbb{E}\left|\frac{1}{L}\sum_{l=1}^L \widetilde{W}_p^p\Big(\theta_l\sharp\mu_n, \theta_l\sharp\nu_m; S_{\theta_l\sharp\mu_n,k_1}, S_{\theta_l\sharp\nu_m,k_2}\Big) - \mathbb{E}_\theta\Big[\widetilde{W}_p^p\Big(\theta\sharp\mu_n, \theta\sharp\nu_m; S_{\theta\sharp\mu_n,k_1}, S_{\theta\sharp\nu_m,k_2}\Big)\Big]\right|^2\right)^{\frac{1}{2}}$$

$$= \left(Var\left[\frac{1}{L}\sum_{l=1}^L \widetilde{W}_p^p\Big(\theta_l\sharp\mu_n, \theta_l\sharp\nu_m; S_{\theta_l\sharp\mu_n,k_1}, S_{\theta_l\sharp\nu_m,k_2}\Big)\right]\right)^{\frac{1}{2}}$$

$$= \frac{1}{\sqrt{L}}Var\left[\widetilde{W}_p^p\Big(\theta\sharp\mu_n, \theta\sharp\nu_m; S_{\theta\sharp\mu_n,k_1}, S_{\theta\sharp\nu_m,k_2}\Big)\right]^{\frac{1}{2}} \leq C_{p,R}\frac{1}{\sqrt{L}},$$

*Table 4.* Approximation errors for fixed $n = 10000$ averaged over 10 runs.

| $k$ | SW (random sampling) | Stream-SW |
|---|---|---|
| 2 | $0.289892 \pm 0.146351$ | $0.0989126 \pm 0.0189911$ |
| 10 | $0.264758 \pm 0.158828$ | $0.0581679 \pm 0.0115344$ |
| 20 | $0.191456 \pm 0.0907318$ | $0.0458331 \pm 0.0142291$ |
| 50 | $0.103948 \pm 0.0582431$ | $0.0404995 \pm 0.0104388$ |
| 100 | $0.110005 \pm 0.0652134$ | $0.0494833 \pm 0.0207168$ |
| 200 | $0.0664362 \pm 0.026191$ | $0.0419366 \pm 0.0155289$ |
| 500 | $0.047502 \pm 0.021997$ | $0.0402404 \pm 0.0156355$ |
| 1000 | $0.0420441 \pm 0.0200953$ | $0.0404598 \pm 0.0131432$ |

for a constant $C_{p,R} > 0$ due to compact support. Combining with Corollary 1

$$\mathbb{E}\left[\left|\widetilde{SW}_p^p(\mu_n, \nu_m; k_1, k_2) - SW_p^p(\mu, \nu)\right|\right] \leq c'_{p,R} \min\{k_1, k_2, \sqrt{n}, \sqrt{m}\}^{-1}, \tag{63}$$

we obtain:

$$\mathbb{E}\left[\left|\widehat{\widetilde{SW}}_p^p(\mu_n, \nu_m; k_1, k_2, L) - SW_p^p(\mu, \nu)\right|\right] \leq c'_{p,R}\left(\min\{k_1, k_2, \sqrt{n}, \sqrt{m}, \sqrt{L}\}^{-1}\right), \tag{64}$$

where $c'_{p,R} > 0$ is a constant depending only on $p$ and $R$.

## B. Additional Materials

**One-sided Stream-SW.** We first state the one-sided version of streaming 1DW and Stream-SW.

**Definition 3.** Given two empirical distributions $\mu_n$ and $\nu_m$ observed in a streaming fashion, and the corresponding quantile sketch $\mathcal{S}_{\mu_n, k_1}$ of $\mu_n$ $k_1 > 1$, one-sided streaming one-dimensional Wasserstein is defined as follow:

$$\widetilde{W}_p^p(\mu_n, \nu_m; \mathcal{S}_{\mu_n, k_1}) = \int_0^1 \left|F_{\mathcal{S}_{\mu_n, k_1}}^{-1}(q) - F_{\nu_m}^{-1}(q)\right|^p \mathrm{d}q. \tag{65}$$

**Definition 4.** Let $k_1 > 1$, and $p \geq 1$. The one-sided *streaming Sliced Wasserstein* (Stream-SW) distance between two empirical distributions $\mu_n \in \mathcal{P}_p(\mathbb{R}^d)$ and $\nu_m \in \mathcal{P}_p(\mathbb{R}^d)$, whose supports are observed in a streaming manner, is defined by

$$\widetilde{SW}_p^p(\mu_n, \nu_m; k_1) = \mathbb{E}_{\theta \sim \sigma}\left[\widetilde{W}_p^p\left(\theta \sharp \mu_n, \theta \sharp \nu_m; S_{\theta \sharp \mu_n, k_1},\right)\right], \tag{66}$$

where $\sigma \in \mathcal{P}(\mathbb{S}^{d-1})$ is a slicing distribution on the unit sphere and $\widetilde{W}_p^p(\theta \sharp \mu_n, \theta \sharp \nu_m; S_{\theta \sharp \mu_n, k_1})$ is defined in Definition 3.

**Gradient approximation.** When using SW as an empirical risk for minimization with respect to a parametric distribution, e.g., $\mu_\phi$, we might be interested in computing the gradient $\nabla_\phi SW_p^p(\mu_\phi, \nu)$ e.g., in point cloud applications and generative modeling applications. We can rewrite $\nabla_\phi SW_p^p(\mu_\phi, \nu) = \nabla_\phi \mathbb{E}_{\theta \sim \sigma(\theta)}\left[\int_{-\infty}^{\infty} |\theta^\top x - F_{\theta \sharp \nu}^{-1}(F_{\mu_\phi}(x))|^p d\mu_\phi(x)\right]$. When $\mu_\phi$ is reparameterizable, i.e., $\mu_\phi = f_\phi \sharp \epsilon$ with $\epsilon$ is a random noise distribution, we can rewrite $SW_p^p(\mu_\phi, \nu) = \mathbb{E}_{\theta \sim \sigma(\theta)}\left[\int_\infty^\infty \nabla_\phi |\theta^\top x - F_{\theta \sharp \nu}^{-1}(F_{\mu_\phi}(f_\phi(z)))|^p d\epsilon(z)\right]$. By using Monte Carlo estimation to approximate expectation and integration, and using streaming approximation for quantile functions (and CDFs), we can form a stochastic estimation of the gradient with respect to $\phi$.

## C. Additional Experiments

**Closed-form for Gaussians.** As we can compute the closed-form of SW in this case, we report the approximation error to the true in Table 4 (varying $k$) and Table 5 (varying $n$) using $L = 10000$. We observe that both Stream-SW and SW with random sampling generally exhibit reduced error as $k$ increases. When $k$ becomes sufficiently large, the difference between

*Table 5.* Approximation errors for fixed $k = 100$ averaged over 10 runs.

| $n$ | SW (random sampling) | Stream-SW |
|---|---|---|
| 500 | $0.112984 \pm 0.0737244$ | $0.0643492 \pm 0.0434308$ |
| 2000 | $0.0504321 \pm 0.0404484$ | $0.0545247 \pm 0.0338965$ |
| 5000 | $0.0869246 \pm 0.0893717$ | $0.0434747 \pm 0.0162144$ |
| 10000 | $0.117947 \pm 0.0717740$ | $0.0389539 \pm 0.0281818$ |
| 20000 | $0.0713203 \pm 0.0467566$ | $0.0368311 \pm 0.0299407$ |
| 50000 | $0.0702557 \pm 0.0622859$ | $0.0355201 \pm 0.0254812$ |

*Table 6.* Approximation errors on MNIST dataset.

| $M$ | $L$ | SW (random sampling) | Stream-SW |
|---|---|---|---|
| 100 | 100 | $0.003467 \pm 0.002563$ | $0.000301 \pm 0.000184$ |
| 1000 | 100 | $0.003720 \pm 0.002776$ | $0.000249 \pm 0.000060$ |
| 100 | 200 | $0.002054 \pm 0.001509$ | $0.000144 \pm 0.000058$ |
| 1000 | 200 | $0.002152 \pm 0.001530$ | $0.000102 \pm 0.000033$ |

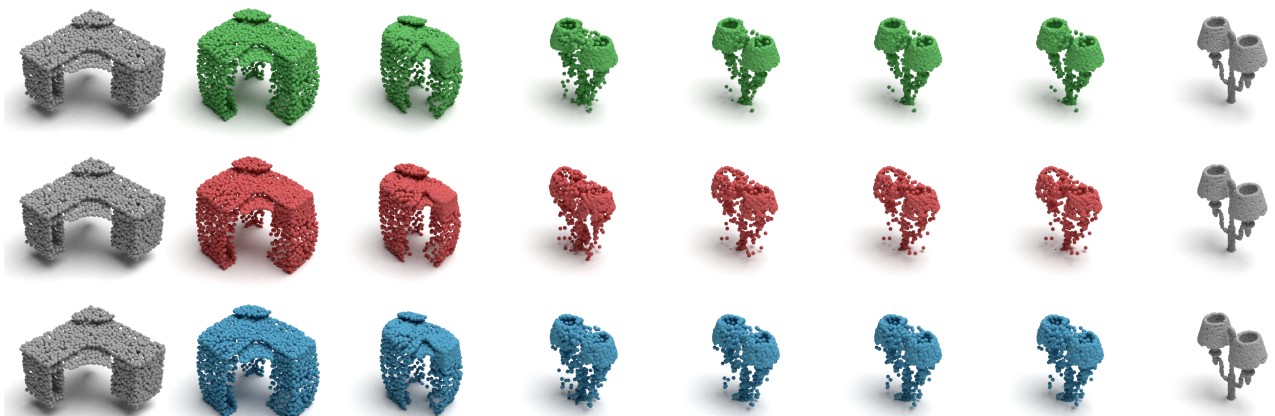

*Figure 6.* Gradient flows from Full SW, SW with random sampling, and Stream-SW in turn ($L = 100, k = 100$).

Stream-SW and SW (random sampling) becomes smaller. Furthermore, increasing $n$ consistently improves the error for Stream-SW, whereas for SW with random sampling, the error decreases in a less predictable manner.

**MNIST Digit.** We compare 5000 MNIST images of "0" with 5000 MNIST images of "1" in 784 dimensions, repeating the experiment 10 times. We show the absolute errors between approximations (Random sampling and Stream-SW) with the full SW distance in Table 6. These results show that Stream-SW achieves much better approximation, and increasing $M$ and $L$ further reduces the error.

**Gradient flows.** As discussed, we visualize flows for ($L = 100, k = 100$) in Figure 6, ($L = 100, k = 200$) in Figure 7, and ($L = 1000, k = 100$) in Figure 8. The qualitative comparison is consistent with the quantitative result in Table 1. We see that increasing $k$ and increasing $L$ leads to better gradient flows in both Wasserstein-2 and perceptual visualization.

**Computational scalability.** We report the computational time in Table 7. We generate empirical distributions from 2D Gaussians. Here, we implement full SW in PyTorch with GPU. SW is very fast when all samples can be kept in memory. However, when the number of atoms increases to 2 million, we are no longer able to compute full SW on the GPU, even though the raw data itself can still be loaded onto the device. For Stream-SW, the computational time increases with $n$, but the growth remains consistent with linear complexity, highlighting the scalability of our algorithm. Moreover, Stream-SW can still be computed for 2 million atoms (and beyond), since the compaction step substantially reduces the number of samples before the final computation. Interestingly, when increasing $k$ from 100 to 1000, the runtime decreases. This is because a larger $k$ results in fewer compaction operations for the KLL sketches. Increasing $k$ further to 10000 leads to only a

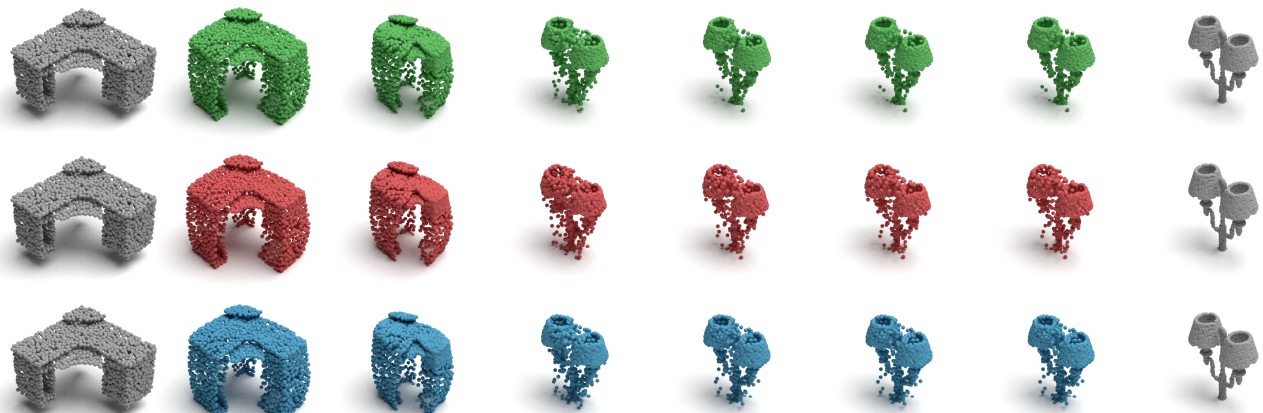

*Figure 7.* Gradient flows from Full SW, SW with random sampling, and Stream-SW in turn ($L = 100, k = 200$).

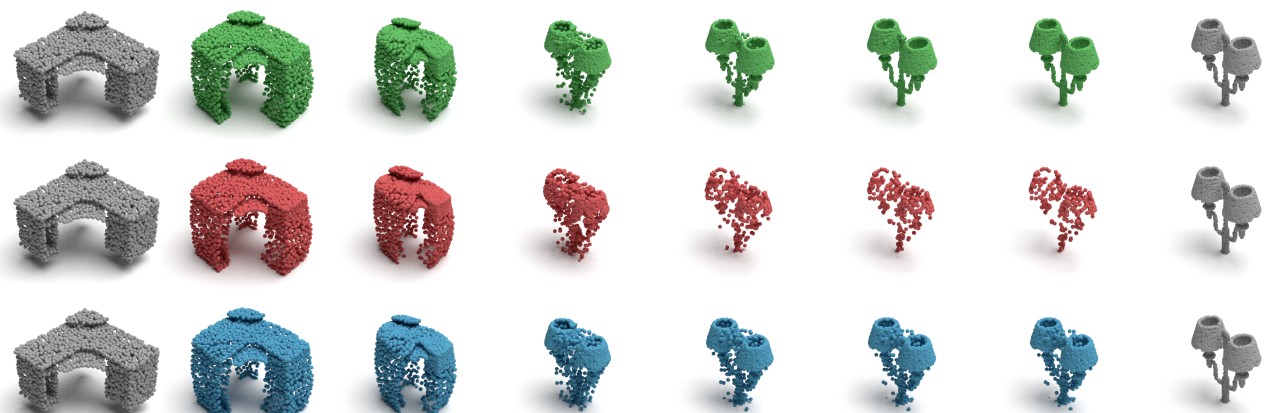

*Figure 8.* Gradient flows from Full SW, SW with random sampling, and Stream-SW in turn ($L = 1000, k = 100$).

*Table 7.* Runtime comparison of full SW, SW with random sampling, and Stream-SW under varying $n$ and $k$.

| $n$ | $k$ | Full SW | SW (random sampling) | Stream-SW |
|---|---|---|---|---|
| 50,000 | 100 | 0.3659 s | 36.7938 s | 10.6451 s |
| 100,000 | 100 | 0.4310 s | 73.2883 s | 20.7061 s |
| 1,000,000 | 100 | 2.7827 s | $\sim$ 12 min | 218.5415 s |
| 2,000,000 | 100 | out of memory | $\sim$ 30 min | 526.7188 s |
| 50,000 | 1,000 | 0.3659 s | 376.9238 s | 9.1104 s |
| 100,000 | 1,000 | 0.4310 s | $\sim$ 100 min | 12.7296 s |
| 1,000,000 | 1,000 | 2.7827 s | too slow | 64.2492 s |
| 2,000,000 | 1,000 | out of memory | too slow | 132.1232 s |
| 50,000 | 10,000 | 0.3659 s | 4200.1544 s | 9.4384 s |
| 100,000 | 10,000 | 0.4310 s | too slow | 14.4618 s |
| 1,000,000 | 10,000 | 2.7827 s | too slow | 67.6551 s |
| 2,000,000 | 10,000 | out of memory | too slow | 133.4081 s |

slight increase in runtime. We also emphasize that the reported runtimes for both Stream-SW and SW with random sampling mainly reflect the streaming computation (i.e., the compaction stage), which is currently implemented in pure Python without optimized libraries or parallelization. Therefore, there remains substantial room for further speed improvements through more efficient implementation.

# D. Computational Infrastructure

We use a HP Omen 25L desktop for conducting experiments.

