# OpenReview forum: "Streaming Sliced Optimal Transport"
_ICML.cc/2026/Conference — ICML 2026 regular_

### Official Review · Reviewer_AKzV · 2026-03-11

**Soundness:** 2
**Presentation:** 1
**Significance:** 4
**Originality:** 3
**Overall Recommendation:** 4
**Confidence:** 3

**Summary:**

Sliced optimal transport (SOT) and the corresponding sliced Wasserstein (SW) distance have recently gained significant attention, as they offer computational advantages over optimal transport and the Wasserstein distance. In parallel, streaming algorithms, which process data streams as a sequence of items, have also become an important area of research, particularly for comparing distributions.

This work focuses on computing the SW distance from streaming samples drawn from two distributions. The SW distance relies on one-dimensional Wasserstein distances between projected probability measures, which admit closed-form expressions in terms of quantile functions. The authors leverage the KKL sketch, a quantile sketch designed to compact data streams. From this sketch, they construct a discrete measure approximating the input measure. Proposition 1 establishes an approximation guarantee for the cumulative distribution function (CDF) obtained from this discrete measure relative to the empirical CDF. Proposition 2 provides a similar guarantee for the corresponding quantile functions. Proposition 3 derives an approximation bound for the one-dimensional Wasserstein distance, both with respect to the empirical distance and the true Wasserstein distance. Propositon 4 studies the convergence of the associated optimal transport maps. Finally, Corollary 1 establishes the approximation error of the streaming SW distance and Theorem 1 extends this result to the case where the SW distance is approximated using Monte Carlo sampling.

Finally, numerical experiments are conducted on Gaussian mixtures and real data, and the method is compared with SW using random sampling.

**Compliance With Llm Reviewing Policy:**

Affirmed.

**Final Justification:**

The paper is technically sound, and the limitations of the numerical experiments were carefully addressed during the rebuttal, where the authors provided additional results. While the initial presentation lacked some rigor and clarity, the authors have committed to improving them in the final version.

Computing the SW distance in a streaming manner is a relevant subject and, to the best of my knowledge, had not been studied before.

The authors were responsive during the rebuttal and addressed all of my concerns. I therefore raised my score to "Weak Accept".

**Key Questions For Authors:**

1. When the two measures receive new points from the stream, the KKL sketch changes: the compactors $C_h$ and weights $w_h$ update. Then the estimated CDF and quantiles will also change and one must recompute from scratch the one-dimensional Wasserstein distances from the quantiles. Isn't this a problem for a streaming algorithm? Shouldn't it be possible to update the distances incrementally using the previous computations?

1. Propositions 1 and 2 show convergence of the streaming CDF and quantile based on $n$ samples towards their empirical counterparts. Why do you not prove convergence towards the true CDF and true quantile of $\mu$, as is done elsewhere in the paper?

1. In Figure 3, the relative error of the SW distance for random sampling increases when $n$ increases. Why is that?

**Limitations:**

Yes

**Strengths And Weaknesses:**

**Soundness :** All proofs were examined but not carefully checked. The numerical experiments are well designed and clearly show the superiority of Stream-SW over SW with random sampling. The computational complexity of Stream-SOT is discussed several times in the paper. However, it would have been useful to include a running time comparison between Stream-SW and SW with random sampling. In addition, comparisons with other sample-based methods (such as measure quantization) would have further strengthened the experimental section.

**Presentation :** In my opinion, the paper suffers from several writing issues. Below is a list of mistakes that I highlighted.

1. The symbol '$=$' can not replace a verb in line 213, left column.
1. The symbol '$\leq$' can not replace a verb in line 184, right column.
1. The end of the sentence line 122, left column, is wrong, also join -> joint.
1. The point '.' is badly placed line 137, left column.
1. The sentence beginning on line 138, left column, shoul be reworded.
1. "that" is missing in the sentence line 151, left column.
1. Line 162, right column, it should be "statistically" and "computationally".
1. The term "as its name" line 200, left column, is wrong.
1. The sentence line 351, right column, seems to be missing a word.
1. "From" should be removed line 770


Some other things could be clarified:

* I found the paragraph "Quantile sketch" (page 4) difficult to follow, particularly for someone who is unfamiliar with quantile sketches. First of all, the authors introduce a set $\{x_1,\cdots,x_k\}$ without specifying the space to which the $x_i$'s belong. It only becomes apparent later, when the items need to be sorted, that these variables are univariate. You should also specify that $w\in\mathbb{R}$. I feel like this paragraph could be more detailed to improve clarity. If the issue is the space constraint, I think that the detailed description of the Gaussian parameters (lines 362L-373L) is uncessary.

* In Propositions 1 and 2, the authors show the convergence of $F_{S_{\mu,k}}$ and $F_{S_{\mu,k}}^{-1}$ towards $F_n$ and $F_n^{-1}$, respectively. The quantities $F_n$ and $F_n^{-1}$ are however never*defined in the main text and only appear in the appendix. Their definitions should be in the main text.

Other than that, the paper is well structured and easy to follow.

**Significance :** Given the current popularity of SOT for comparing distributions, computing the SW distance in a streaming setting is a relevant problem which was well addressed in this paper.

**Originality :** This work provides a method for computing the SW distance between probability measures, available in a streaming setting. Computing Wasserstein distances between distributions from streaming samples has been addressed in the entropic case, but not in the sliced framework. This work clearly feels this gap.

---

> ### Author Rebuttal · Authors · 2026-03-28
>
> We would like to thank the reviewer for spending time reviewing our work and providing valuable feedback. We would like to extend our discussion as follows:
>
> **Q24** ... running time comparison between Stream-SW and SW with random sampling. In addition, comparisons with other sample-based methods...
>
> **A24** We observe that Stream-SW and SW with random sampling have similar computation times when random sampling is implemented in a streaming fashion (eservoir sampling.) For $n=5000, k=100$, Stream-SW takes 3.42s vs. SW (random sampling with fair implementation) 3.45s; for $n=10000, k=100$, 7.31s vs. 7.40s. The slight advantage of Stream-SW may be due to random sampling requiring multiple acceptance/rejection steps. Other types of sampling could also be used; however, if sampling is only used to satisfy the memory constraint, it does not guarantee the same approximation quality as Stream-SW.
>
> Regarding measure quantization, to the best of our knowledge, it is applicable only when the full set of samples is available, and we are not aware of a streaming variant (supporting online updates under memory constraints). Even if such a method existed, establishing theoretical guarantees would be challenging. In contrast, Stream-SW has solid theoretical support. Additionally, Stream-SW can be interpreted as a form of measure quantization, since the inverse Radon transform can reconstruct the original measure from the projected quantiles, though this reconstruction can be computationally expensive in high dimensions. As our focus is on sliced OT, we leave this application for future work.
>
> **Q25** List of writing mistakes.
>
> **A25** We have corrected all identified writing mistakes in the revised manuscript.
>
> **Q26** The authors introduce a set $x_1,\ldots,x_n$ without specifying the space to which the $x_i$ belong..
>
> **A26** Thank you for pointing this out. The set $x_1,\ldots,x_n$ belongs to the real line, i.e., $x_i \in \mathbb{R}$. The weights $w$ are positive numbers, i.e., $w \in \mathbb{R}_+$. We will revise the paragraph accordingly to improve clarity.
>
> **Q27** In Propositions 1 and 2 ... However, the quantities are never defined in the main text and appear only in the appendix. Their definitions should be included in the main text.
>
> **A27** We will move the definitions of the quantile functions and cumulative distribution functions to the main text for clarity.
>
> **Q28** When two measures receive new points from the stream, the KKL sketch changes: the compactors and weights are updated... Isn’t this a problem for a streaming algorithm? Could distances be updated incrementally using previous computations?
>
> **A28** Thank you for the insightful question. When the compactors and weights update, the discrete empirical measures from the sketch change. Because the Wasserstein distance relies on optimal transport, removing a single atom affects the entire matching, so recomputation is necessary. However, the most time-consuming step i.e., sorting the atoms/supports, occurs during compaction. The realignment step then consumes only linear time with respect to the sketch size. Exact computation is important for theoretical development, as it provides the precise distance and allows use of the triangle inequality. Incremental updates of the distance (based solely on removed points) are possible but non-trivial, so we leave this as future work.
>
> **Q29** Propositions 1 and 2 show convergence of streaming CDFs and quantiles based on samples toward their empirical counterparts. Why not prove convergence toward the true CDF and quantiles, as done elsewhere?
>
> **A29** Convergence toward the true CDF is implicitly established in Proposition 3, where we analyze the convergence of the streaming 1D Wasserstein distance to the population 1D Wasserstein distance. This reduces to bounding the error between the streaming quantile estimate and the true population quantile, which can be decomposed via a triangle inequality into: (i) the error of the streaming estimate versus the empirical quantile (Proposition 2), and (ii) the well-known error of the empirical quantile versus the true quantile. Thus, the result requested by the reviewer corresponds to Proposition 3.
>
> **Q30** In Figure 3, the relative error of the SW distance for random sampling increases with $n$. Why?
>
> **A30**  The relative error is measured as the absolute difference between the estimated SW (Stream-SW or SW with random sampling) and the true SW for empirical samples, $|\cdot - SW_p(\mu_n, \nu_n)|$. For random sampling, we sample $3k + 2 \log(n/(2k/3))$ points. As $n$ increases, the number of sampled points grows slowly relative to $n$, so the relative error increases because the sample fraction decreases. In contrast, Stream-SW updates KKL sketches and projection quantiles to retain order statistics for any $n$, so the approximation quality remains approximately constant as $n$ increases.

---

> > ### Author Rebuttal · Reviewer_AKzV · 2026-04-02
> >
> > I want to thank the authors for taking the time to answer my questions in detail.
> >
> > Regarding measure quantization, which boils down to k-means, there is a line of work on streaming k-means which operates under online updates and memory constraints, see e.g. [R1].
> >
> > I am still not entirely convinced of Figure 3, which I find difficult to interpret. I feel like a more informative figure would be to compare the estimators directly to $SW(\mu,\nu)$ in a controlled setting. For instance, you could consider Gaussian measures, for which 1D projected Wasserstein distance has a closed form, and approximate SW using high-precision Monte-Carlo integration over directions. I believe this would provide a clearer comparison between Stream-SW and SW with random sampling.
> >
> > Otherwise, all my other concerns have been addressed, and I raise my score to 4 (weak accept).
> >
> > [R1] Zhang, Y., Tangwongsan, K. and Tirthapura, S., 2017, April. Streaming k-means clustering with fast queries. In 2017 IEEE 33rd international conference on data engineering (ICDE) (pp. 449-460). IEEE.

---

> > > ### Author Response · Authors · 2026-04-02
> > >
> > > Thank you very much increasing the score to 4 and for asking additional questions.
> > >
> > > # Streaming K-means
> > >
> > > Thank you for pointing us to the literature on streaming k-means. We agree that this is a highly relevant and promising direction for distributions comparison in the streaming setting. Unfortunately, because the referenced work does not provide readily available code and because of our limited expertise in this specific area, we are not able to include an empirical comparison with that approach within the short rebuttal timeline. Nevertheless, we would like to provide the following discussion.
> > >
> > > First, the KKL sketch may not be the optimal sketching algorithm for streaming sliced Wasserstein (SW). We adopt the KKL sketch because it comes with theoretical guarantees for CDF approximation, which can then be extended to quantile approximation and subsequently connected to the theory of sliced optimal transport, including both transportation cost and transport maps. As an initial step toward connecting sliced optimal transport with streaming algorithms, our current Stream-SW framework is likely not yet optimal, since the KKL sketch was originally designed for CDF approximation rather than being specifically tailored to 1D Wasserstein estimation.
> > >
> > > Second, we agree that combining streaming measure quantization methods, such as streaming k-means, offers another natural route for defining streaming sliced optimal transport. However, establishing theoretical guarantees for the resulting approximation of sliced Wasserstein distance is nontrivial. Moreover, to the best of our knowledge, the streaming k-means approach in [R1] does not support automatically adaptive growth of the coreset size, in contrast to the KKL sketch.  That said, we view this more as a limitation of the streaming k-means formulation in [R1] rather than a fundamental obstacle. We believe it is entirely possible to extend streaming k-means ideas to the SW setting, for example through **global quantization** (quantizing the original measures) or **projection quantization** (quantizing the projected 1D measures). Exploring these directions, however, involves substantial additional technical development of algorithms and theory and is beyond the scope of the present work. We therefore leave this as an important direction for future research.
> > >
> > > Overall, we hope the reviewer agrees that designing streaming algorithms for sliced optimal transport is a novel and promising research direction. In particular, it leverages several appealing structural properties of sliced optimal transport, such as quantile-function-based representations, sorting-based computation, and the interaction across projections.
> > >
> > > # Figure 3
> > >
> > > For Figure 3, we have changed the evaluation setting as suggested. We compare two Gaussians (in 2D)  and utilize the closed-form solution of projected Gaussians with 10000 projections as the true. For fixed $n=10000$, the following results are averaged over 10 runs:
> > >
> > > | k | Stream-SW (mean ± std) | SW (random sampling) (mean ± std) |
> > > |-------|----------------------------|----------------------|
> > > | 2 | 0.0989126 ± 0.0189911 | 0.289892 ± 0.146351 |
> > > | 10 | 0.0581679 ± 0.0115344 | 0.264758 ± 0.158828 |
> > > | 20 | 0.0458331 ± 0.0142291 | 0.191456 ± 0.0907318 |
> > > | 50 | 0.0404995 ± 0.0104388 | 0.103948 ± 0.0582431 |
> > > | 100 | 0.0494833 ± 0.0207168 | 0.110005 ± 0.0652134 |
> > > | 200 | 0.0419366 ± 0.0155289 | 0.0664362 ± 0.026191 |
> > > | 500 | 0.0402404 ± 0.0156355 | 0.047502 ± 0.021997 |
> > > | 1000 | 0.0404598 ± 0.0131432 | 0.0420441 ± 0.0200953|
> > >
> > >
> > >
> > > For fixed $k=100$, the following results are averaged over 10 runs:
> > >
> > > | n | Stream-SW (mean ± std) | SW (random sampling) (mean ± std) |
> > > |-------|----------------------------|----------------------|
> > > | 500 | 0.0643492 ± 0.0434308 | 0.112984 ± 0.0737244 |
> > > | 2000 | 0.0545247 ± 0.0338965 | 0.0504321 ± 0.0404484 |
> > > | 5000 | 0.0434747 ± 0.0162144 | 0.0869246 ± 0.0893717 |
> > > | 10000 | 0.0389539 ± 0.0281818 | 0.117947 ± 0.0717740 |
> > > | 20000 | 0.0368311 ± 0.0299407 | 0.0713203 ± 0.0467566 |
> > > | 50000 | 0.0355201 ± 0.0254812 | 0.0702557 ± 0.0622859 |
> > >
> > > We observe that both Stream-SW and SW with random sampling generally exhibit reduced error as $k$ increases. When $k$ becomes sufficiently large, the difference between Stream-SW and SW (random sampling) becomes smaller. Furthermore, increasing $n$ consistently improves the error for Stream-SW, whereas for SW with random sampling, the error decreases in a less predictable manner. We will include this additional Gaussian case in the revised version. We are happy to take any further questions

---

### Official Review · Reviewer_dJpM · 2026-03-12

**Soundness:** 3
**Presentation:** 4
**Significance:** 3
**Originality:** 3
**Overall Recommendation:** 5
**Confidence:** 2

**Summary:**

This paper introduces Streaming Sliced Wasserstein (Stream-SW), a novel algorithm that directly and efficiently estimates the sliced Wasserstein distance from continuous sample streams.  To alleviate computational and memory issues, the method makes use of quantile sketching techniques to dynamically approximate cumulative distribution functions and quantile functions with a minimal, logarithmic memory footprint. This enables the construction of a highly efficient streaming estimator for the one-dimensional Wasserstein distance (1DW), which can be easily extended to high-dimensional distributions. Supported by rigorous theoretical guarantees on approximation error, Stream-SW significantly outperforms traditional random sub-sampling in accuracy and efficiency. Its practical utility is demonstrated across various downstream tasks, including point cloud classification, gradient flows, and streaming change point detection, making it a highly scalable tool for memory-constrained machine learning applications.

**Compliance With Llm Reviewing Policy:**

Affirmed.

**Final Justification:**

This paper is solid, and the authors solved my concerns during the rebuttal period. I will maintain my current given score.

**Key Questions For Authors:**

1. For truly high-dimensional data (e.g., $d \ge 512$), does the required number of projections $L$ need to increase significantly to maintain accuracy? Furthermore, how does Stream-SW empirically perform in such high-dimensional settings?
2. How does Stream-SW compare to more advanced online OT estimators regarding the efficiency-accuracy trade-off, particularly in terms of computational speedup?
3. In line 1035 of the appendix, you mention "keeping n = 1000." Is this a typo? Should it be "10000" or something?

**Limitations:**

While the paper is well-executed, the evaluation primarily focuses on moderate dimensions and basic baselines. Exploring the method's behavior in much higher-dimensional spaces and its positioning against a broader range of advanced streaming estimators would further clarify its scalability and unique efficiency-accuracy trade-offs.

**Strengths And Weaknesses:**

Strengths:
1. Novel Contribution: The paper introduces the first streaming estimators for the 1D Wasserstein distance and the corresponding 1D optimal transport map. By elegantly applying this to all projections, it derives Stream-SW, making it a highly novel approach.
2. Solid Theory & Rigorous Bounds: The paper provides a strong theoretical foundation, featuring rigorous proofs, strict bounds, and a comprehensive trade-off analysis.
3. Strong Empirical Results: The experimental validation is thorough, convincingly demonstrating that Stream-SW achieves highly effective performance in practice.


Weaknesses:
1. Although Stream-SW outperforms basic methods (Sliding Window, Rand-SW), adding comparisons with more advanced streaming baselines would further strengthen the paper by clearly highlighting its unique advantages and efficiency-accuracy trade-offs.

---

> ### Author Rebuttal · Authors · 2026-03-28
>
> We would like to thank the reviewer for the time and constructive feedback. We would like to extend our discussion as follows:
>
> **Q20** Although Stream-SW outperforms basic methods (Sliding Window, Rand-SW), adding comparisons with more advanced streaming baselines would further strengthen the paper by clearly highlighting its unique advantages and efficiency-accuracy trade-offs.
>
> **A20:**   To the best of our knowledge, Stream-SW is the first streaming estimator for the Sliced Wasserstein (SW) distance. To demonstrate that Stream-SW provides correct SW estimates from sample streams, we compare it with natural adaptations of SW to streaming samples, such as Sliding Window and Rand-SW.
>
>  **Q21**  For truly high-dimensional data (e.g., $d \ge 512$), does the required number of projections $L$ need to increase significantly to maintain accuracy? Furthermore, how does Stream-SW empirically perform in such high-dimensional settings?
>
> **A21:**   Increasing $L$ improves accuracy for both SW and Stream-SW. From Theorem 1, the estimation accuracy of Stream-SW improves as $L$ increases. For measures with compact support, this accuracy depends on the support diameter rather than the dimensionality (as a constant for the bound). For structured measures, accuracy often depends on the variance of 1D projected Wasserstein distances, which can implicitly depend on the dimension when the measure has varying structure across dimensions [6]. This is reflected in Table 1 in our experiments.
>
> From the literature, it is generally known that the number of projections should roughly match the data dimension for uniform sampling of directions. To empirically verify this, we have added an additional experiment. In particular, we have compared 5000 MNIST images of “0” with 5000 MNIST images of “1” in 784 dimensions, repeating the experiment 10 times:
>
> | $L$    | $k$   | Stream-SW                  | SW (random sampling)         |
> |--------|-------|----------------------------|------------------------------|
> | 100    | 100   | 0.000301 ± 0.000184       | 0.003467 ± 0.002563          |
> | 1000   | 100   | 0.000249 ± 0.000060       | 0.003720 ± 0.002776          |
> | 100    | 200   | 0.000144 ± 0.000058       | 0.002054 ± 0.001509          |
> | 1000   | 200   | 0.000102 ± 0.000033       | 0.002152 ± 0.001530          |
>
> These results show that Stream-SW achieves much better approximation, and increasing $L$ and $k$ further reduces error.
>
> We would like to recall that other projection selection strategies can be applied [7] to Stream-SW to make it projection efficient, but this is orthogonal to our focus on streaming estimation, which aims to efficiently estimate Wasserstein distances per slice. Therefore, we leave the investigation of projection selection for Stream-SW for future work, especially since the current paper is already dense as mentioned by the reviewers.
>
> [6] Nadjahi et al., *Statistical and Topological Properties of Sliced Probability Divergences*, NeurIPS 2020.
> [7] Nguyen et al., *Energy-Based Sliced Wasserstein Distance*, NeurIPS 2023.
>
> **Q22** How does Stream-SW compare to more advanced online OT estimators regarding the efficiency-accuracy trade-off, particularly in terms of computational speedup?
>
> **A22:**  Sliced Wasserstein (SW) and Stream-SW are alternatives to, rather than replacements for, Wasserstein distances. SW and Stream-SW have sample complexities that do not depend on dimension, unlike standard Wasserstein. The only existing online Wasserstein estimator, online Sinkhorn, has time complexity $\mathcal{O}(n^2)$ and memory complexity $\mathcal{O}(n)$, since it requires storing all previously seen data.  In contrast, Stream-SW has time complexity $\mathcal{O}((n/k) \log(n/k))$ for initial sketch size $k$, and memory complexity $\mathcal{O}(\log(n/k))$, clearly offering substantial computational savings.
>
> We note the distinction between online and streaming algorithms: online algorithms have no memory constraints, whereas streaming algorithms operate under strict memory limits. Stream-SW is specifically designed for the streaming setting, supporting online updates while respecting memory restrictions, making it unique in the current literature.
>
> **Q23** In line 1035 of the appendix, you mention "keeping n = 1000." Is this a typo? Should it be "10000" or something?
>
> **A23:**   Yes, this is a typo. We have corrected it in the revision.

---

> > ### Author Rebuttal · Reviewer_dJpM · 2026-04-03
> >
> > My concerns have been addressed, and I believe I have given a fair score for this paper. Best of luck!

---

> > > ### Author Response · Authors · 2026-04-03
> > >
> > > Thank you very much for the acknowledgement! Please feel free to ask if any further questions arise!

---

### Official Review · Reviewer_7SAZ · 2026-03-16

**Soundness:** 3
**Presentation:** 1
**Significance:** 3
**Originality:** 3
**Overall Recommendation:** 5
**Confidence:** 3

**Summary:**

The work is dedicated to sliced Wasserstein (SW) distance between probability distributions. Specifically, the authors develop an online algorithm for estimating SW from streams of data that doesn’t require storing the full dataset. Since SW between two distributions is expressed via 1-dimensional Wasserstein (1DW) distances between their projections, the work leverages the fact that 1DW is the integral of the absolute difference (to the power $p$) of quantile functions. The authors employ a data structure called quantile sketch which allows to estimate quantile functions in an online fashion, keeping only a subset of samples. The proposed method (called Stream-SW) therefore approximates SW based on the estimated quantiles of projections. In the case of compactly supported measures. the authors bound the error in terms of the sketches’ sizes, the total numbers of samples from the streams, and the number of projections. Moreover, the space complexity bound is derived. Extensive experiments showcase the algorithm’s performance.

**Compliance With Llm Reviewing Policy:**

Affirmed.

**Final Justification:**

The rebuttal thoroughly addressed my concerns, mostly related to presentation and clarity. The authors clarified the baseline description, added limitations, and promised to fix other presentation issues. Given these corrections, I raised my score from 4 to 5 (accept). The work remains technically solid and fills the gap by developing the first online method for SW with rigorous complexity bounds.

**Key Questions For Authors:**

1. Left column, L20-22: did you mean “*the integral of* the absolute difference…”?
1. Left column, L86-87: does $O(m^{-1/d})$ indicate the convergence rate, rather than the sample complexity (i.e., the number of samples required)?
1. The contributions paragraph is a bit too detailed, consider making it a little shorter and more to the point.
1. Left column, L126: $N$ -> $n$
1. Right column, L134: pdf -> PDF. Besides, add the full name “Probability Density Function”, since you are using the acronym for the first time here.
1. Right column, L146: Equation 4 -> Equation (4)
1. Left column, L213: do you need to introduce $c$? Can you simply plug $c=2/3$ directly into the formula to make it clearer?
1. Left column, L214-218: By the total weight, do you mean $\sum_{h=1}^H w_h k_h$? Why is it equal $n$, i.e., the total number of samples in the stream (might be unknown)? Are the points discarded when $C_H$ reaches capacity? Please revise the description to make it clearer.
1. Proposition 1: do you assume $n>k$? If yes, please mention that here and in subsequent statements.
1. Equation (8): $F_n$ wasn’t defined.
1. Equation (12): should the integral be over $[0,1]$?
1. Left column, L261, second summation: $n$ -> $m$
1. Equation (13): does $\widetilde{W}_p^p$ depend on $\mu_n, \nu_m$ only through the sketches? If yes, the first two arguments are redundant.
1. Right column, L328: can we simply set $k:=\max(k_1, k_2)$?
1. Left column, L362-372: please use math mode instead of inline equations here.
1. Section 4: describe the baseline (SW with subsampling) in more detail. Is my understanding correct that you collect all samples, retain a random subset of the specified size (sampling without replacement), and compute SW in an offline fashion from this subset? Also, please explicitly write that you use $m=n$ and $k_1=k_2$ (denoted $k$), if that is the case.
1. Figure 3: consider using logarithmic Y-scale for the relative error. Also, please mathematically define the relative error in text.

**Limitations:**

Limitations should be discussed

**Strengths And Weaknesses:**

Strengths:
1. To the best of my knowledge, this is the first **online** method for SW estimation.
1. Time, space and sample complexity bounds are derived.
1. The method performs well in diverse experiments.

Weaknesses (see the questions section for more details):
1. The paper has many typos, language mistakes, and poorly formulated sentences, e.g., left column, lines 26-28, 122-124, 136-138, 151-152.
1. The quality of the presentation should be significantly improved.
1. Experiments should be described in greater detail.

---

> ### Author Rebuttal · Authors · 2026-03-28
>
> We thank the reviewer for their time and valuable feedback on our presentation. We have corrected all typos and revised the paper based on the reviewer’s suggestions. Our responses to the questions are as follows:
>
> **Q4** Left column, L20-22: did you mean “the integral of the absolute difference…”?
>
> **A4** Yes, we refer to the function norm.
>
> **Q5** Left column, L86-87: does $\mathcal{O}(m^{-1/d})$ indicate the convergence rate, rather than the sample complexity (i.e., the number of samples required)?
>
> **A5** Correct, we mean the convergence/compression rate. We will remove "sample complexity" to avoid missunderstanding.
>
> **Q6** The contributions paragraph is a bit too detailed, consider making it shorter and more to the point.
>
> **A6** We have shortened it for clarity.
>
> **Q7** "Left column, L126: $N \to n$"; "Right column, L134: pdf -> PDF. Besides, add the full name “Probability Density Function”, since you are using the acronym for the first time here."; "Right column, L146: Equation 4 -> Equation (4)".
>
> **A7** We have fixed the typos.
>
> **Q8** Left column, L213: do you need to introduce $c$? Can you plug $c=2/3$ directly into the formula to make it clearer?
>
> **A8** We introduce $c$ to control hierarchical size, but we will plug it directly into the formula to improve clarity as suggested.
>
> **Q9** Left column, L214-218: By the total weight, do you mean  $\sum_{h=1}^H w_h k_h$? Why is it equal  $n$, i.e., the total number of samples in the stream (might be unknown)? Are the points discarded when  $C_H$ reaches capacity? Please revise the description to make it clearer.
>
> **A9** We mean $\sum_{h=1}^h w_h = n$ because during compaction, KKL removes an even number of samples and doubles the weights of the remaining ones, ensuring the total sum remains $n$ (number of observed samples). We will revise the paragraph for clarity.
>
> **Q10** Proposition 1: do you assume  $n>k$? If yes, please mention that here and in subsequent statements.
>
> **A10** Yes, we will explicitly state this assumption in the revised text.
>
> **Q11** Equation (8): $F_n$ wasn’t defined.
>
> **A11** It is the inverse CDF (quantile function), currently defined in Appendix A.2. We will move the definition to the main text.
>
> **Q12** Equation (12): should the integral be over  $[0,1]$?
>
> **A12** Yes, it was a typo. The proof uses the correct integral, and we will fix it in the main text.
>
> **Q13** Left column, L261, second summation: $n \to m$
>
> **A13** We have fixed the typo.
>
> **Q14** Equation (13): does  $\widetilde{W}_p^p$ depend on  $\mu_n,\nu_m$ only through the sketches? If yes, the first two arguments are redundant.
>
> **A14** We agree. However, we may remove the second two arguments as we want to highlight that it is a function of the two empirical measures.
>
> **Q15** Right column, L328: can we simply set $k:=\max(k_1,k_2)$?
>
> **A15** Yes, we will do so in the revision.
>
> **Q16** Left column, L362-372: use math mode instead of inline equations.
>
> **A16** We have revised the paper accordingly.
>
> **Q17** Section 4: describe the baseline (SW with subsampling) in more detail. Is my understanding correct that you collect all samples, retain a random subset of the specified size (sampling without replacement), and compute SW in an offline fashion from this subset? Also, please explicitly write that you use  $m=n$   and $k_1=k_2$ (denoted  $k$), if that is the case.
>
> **A17**A streaming scheme to obtain $K$ uniform samples with replacement maintains $K$ slots updated independently as new samples arrive. After observing the $t$-th sample $x_t$, each slot replaces its content with $x_t$ with probability $1/t$, otherwise unchanged. Each slot acts as an independent size-one reservoir sampler. After $t$ observations, each slot contains a uniform sample from ${x_1, ..., x_t}$, yielding $K$ i.i.d. draws. After that, compute SW  from this subset. In simulations, since all samples are in memory, we perform uniform sampling directly. We will add this discussion to the revision of the paper and  explicitly write that we use $m=n$   and $k_1=k_2$ (denoted  $k$) .
>
>
> **Q18** Figure 3: consider logarithmic Y-scale and define relative error. Also, please mathematically define the relative error in text.
>
> **A18** Relative error is the absolute difference between the estimated SW (Stream-SW or SW with random sampling) and the true SW: $|\cdot - SW_p(\mu_n, \nu_m)|$. We will revise the figure and text accordingly.
>
> **Q19** Limitations should be discussed.
>
> **A19** Current limitations include: Stream-SW implementation not fully optimized, statistical bounds may not be sharp, and KKL sketch may not be optimal for Stream-SW. We will add this discussion.

---

> > ### Author Rebuttal · Reviewer_7SAZ · 2026-04-01
> >
> > My questions have been addressed and I am raising the score by one point.

---

> > > ### Author Response · Authors · 2026-04-01
> > >
> > > Thank you very much for your acknowledgment and for increasing the score to 5. Please feel free to ask if any further questions arise.

---

### Official Review · Reviewer_dXZ7 · 2026-03-20

**Soundness:** 3
**Presentation:** 2
**Significance:** 2
**Originality:** 3
**Overall Recommendation:** 3
**Confidence:** 4

**Summary:**

This article proposes to apply the Karnin, Lang, and Liberty contribution on optimal quantile computation in streams to the computation of sliced Wasserstein distance. The computational complexity of sliced Wasserstein is essentially O(L n log(n)) where L is the number of projections along sampled directions. In the context of a large stream of data and reduced computing/memory resources, the proposed method reduces the computational burden to essentially O( L k log k + L (n/k) log n/k) where k is the parameter defining the memory size of the method which is in k + log(n/k).
The article gives some theoretical properties of the method in particular a (probabilistic) exponential rate of convergence of the streamed estimator to the true value in the case of a compact domain.
Experiments are shown to demonstrate the feasibility of the method, in the case of mixture of Gaussians, demonstrating the practical potential of the method in the non-compact case, on gradient flows, point cloud classification and change point detection.

**Compliance With Llm Reviewing Policy:**

Affirmed.

**Final Justification:**

In view of the additional runtime results, I think that these results should have been included in the first submitted version of the paper, and could have changed the overall opinion of the reviewers. I increase the score by 1 since I acknowledge that the method has the potential to be used in certain settings.

**Key Questions For Authors:**

The key point for this algorithm is the computational time associated with the sketch computation. This should be thoroughly discussed in the paper.
I am willing to raise my score after the rebuttal if my concern is clearly addressed.

**Limitations:**

yes.

**Strengths And Weaknesses:**

Exploring the possibility of fixed-sized memory for computing the sliced Wasserstein distance is meaningful in the context of very large empirical measures, which are by now common.
The sketching of the data is done by the use of the optimal KLL scheme, which actually applies to cumulative distribution function (cdf) but not directly on the inverse of the cdf. Although the result of KLL applies to the cdf, the convergence results are derived from the use of the inverse function on the sketch provided by KLL. In particular, the constant C_p,R obtained in Proposition 3 could be potentially worse than what would have been given by a direct treatment of the inverse of the cdf.


The presentation is clear, although I had to go to the KLL paper to be able to implement the stream SW. I think more emphasis on this method could be of interest for the reader. Also, it would be informative to explain how the stream method can be applied to online for the SW computation in the explanation of the KLL algorithm.
This work provides a nice adaptation of an existing important result by KLL to the sliced Wasserstein distance computation and show that one can obtain accurate estimation of the SW distance with a low memory footprint.

However, I have an important concern with this article. I used the code provided in the supplementary material to compute the streamed SW distance, and it appears that the computational time necessary to update the sketch on my cpu laptop is roughly 8 times more than to compute the true SW distance on 100 000 points (without any reduction, full batch here). I understand that the theoretical complexity of the algorithm is fairly lower, and the efficiency of the algorithm is certainly very dependent on the data structures and the implementation. However, in its current form the code proposed is severely slower than standard SW and I wonder how slow it can be when scaled to a very large number of points, which is the targeted situation.
For that reason, I think adding a section on experiments on compute time is also crucial, and discussing the issue in the paper is needed.

---

> ### Author Rebuttal · Authors · 2026-03-28
>
> We would like to thank the reviewer for the time and constructive feedback. We address the comments as follows:
>
> **Q1** The constant $C_{p,R}$ in Proposition 3 could be potentially worse...
>
> **A1** The constant $C_{p,R}$ depends on the diameter of the supports $R$. To our knowledge, this is the best bound without additional structure on the measures. In practice, many cases have bounded $R$, e.g., 3D point clouds or normalized images, making the bound practical. Similar constants appear in other statistical results for Wasserstein and sliced Wasserstein (SW) [1,2]. While sharper bounds can be achieved by adding more structures, our main goal is to introduce Stream-SW and provide a first theoretical and algorithmic overview.
>
> [1] Nietert et al., Statistical Robustness and Computational Guarantees for Sliced Wasserstein Distances, NeurIPS 2022.
>
> [2] Genevay et al., Sample Complexity of Sinkhorn Divergences, AISTATS 2019.
>
> **Q2** ... it would be informative to explain how the stream method can be applied to online for the SW computation in the explanation of the KLL algorithm.
>
> **A2** Due to the 8-page limit and our new results connecting KLL sketch to 1D OT and SW, we kept the KLL discussion concise. In the revision, we will add pseudo-algorithms for KLL sketch and Stream-SW, and discuss how 1D Wasserstein distances can be computed from sketches (Algorithm 3 in [3]).
>
> [3] Séjourné et al., Faster Unbalanced Optimal Transport: Translation Invariant Sinkhorn and 1-D Frank-Wolfe, AISTATS 2022.
>
> **Q3** Streamed SW in the supplementary code appears ~8× slower than full SW on 100,000 points. How does this scale to larger data? A compute-time discussion is needed.
>
> **A3**  Thank you for testing and verifying our code. To clarify the computational aspects of Stream-SW, it is helpful to consider two scenarios: (1) streaming samples with limited memory, and (2) full samples with sufficient memory to compute the standard SW.
>
> First, Stream-SW is designed for scenario (1), where we cannot compute SW on the full sample set due to memory constraints. For example, with $n = 100000$ samples but memory for only 100, full SW is infeasible, so some form of sampling is required. A simple streaming approach is to maintain $K$ slots and update them independently as new observations arrive. Specifically, after observing the $t$-th sample $x_t$, each slot $k = 1, \dots, K$ replaces its current value with $x_t$ with probability $1/t$, otherwise it remains unchanged. Equivalently, each slot acts as an independent size-one reservoir sampler, so the $K$ stored values are i.i.d. draws from the empirical distribution of the first $t$ samples, yielding uniform sampling with replacement.
>
> Stream-SW improves upon this approach by leveraging quantile approximations, providing better accuracy and memory efficiency. In native Python, for $n=5000, k=100$, Stream-SW takes 3.42s vs. SW (random sampling with fair implementation) 3.45s; for $n=10000, k=100$, 7.31s vs. 7.40s. While other sampling methods can be used, they lack principled guarantees like Stream-SW. We use full SW in the paper only as ground truth to evaluate Stream-SW. Overall, the primary aim of Stream-SW aligns with standard streaming algorithms [4,5], where memory  is the main constraint.
>
> For scenario (2), we do not claim Stream-SW is faster than SW. Theoretical complexity is near-linear in $n$, similar to SW. Efficiency gains depend on the sketch size $k$, larger $k$ reduces the number of compactions. In practice, $n$ must be very large to observe substantial speed benefits. For the implementations differ: SW uses optimized PyTorch with fully vectorized projections, sorting, and 1D Wasserstein computation, with all samples in memory. Stream-SW uses KLL sketch in native Python without parallelization; each projection is updated point by point. Additionally, SW computes 1D Wasserstein between empirical measures with the same number of supports (just sorting), whereas Stream-SW requires discrete measures with more complex operations (e.g., Algorithm 3 in [3]). These factors explain why Stream-SW appears slower than full SW. Using lower-level languages (e.g., C) and parallelization could reduce this gap. Importantly, Stream-SW is not intended as a replacement for SW in this setting; its focus is on streaming estimation under memory constraints.
>
> Looking forward, Stream-SW may benefit scenarios involving accelerated devices (e.g., GPUs) with limited memory i.e., streaming algorithms allow memory-efficient processing, making Stream-SW a suitable candidate for such applications.
> Given these points, we hope the reviewer will reconsider the evaluation, as Stream-SW is a novel direction with potential broad applications in distributed and memory-constrained environments. We will add this discussion to the revision.
>
> [4] Karnin et al., Optimal Quantile Approximation in Streams, FOCS.
> [5] Felber et al., A Randomized Online Quantile Summary in O((1/ε) log(1/ε)) Words, APPROX/RANDOM 2015.

---

> > ### Author Rebuttal · Reviewer_dXZ7 · 2026-04-01
> >
> > I thank the authors for their answers. I acknowledge that the method is novel and further accelerations can improved this proposal. My main concern is partially addressed.
> >
> > Could the author provide a table of the time performance of the algorithm varying n, k where n goes large, say 10^6 ?

---

> > > ### Author Response · Authors · 2026-04-01
> > >
> > > Thank you very much for your prompt acknowledgment. We would like to provide the requested runtime performance of the algorithms. In our setup, we fix the number of projections at 100 and generate
> > > $n$ samples from the same 2D Gaussian mixture used in the paper. Data generation is performed on the CPU, after which the samples are transferred to the GPU for compaction and subsequent computation of the sliced Wasserstein distance and the one-dimensional Wasserstein distances. We vary $n \in \\{ 50,000; 100,000; 1,000,000;2,000,000\\} $ and $k \in \\{100;1,000; 10,000\\}$. We observe the following result:
> > >
> > > | n        | k   | Stream-SW  | SW  (random sampling) (s) | Full SW |
> > > |----------|-----|------------------|--------------------|-------------|
> > > | 50,000   | 100 | 10.6451    (s)      | 36.7938   (s)          | 0.3659 (s)      |
> > > | 100,000  | 100 | 20.7061    (s)       | 73.2883   (s)          | 0.4310   (s)    |
> > > | 1,000,000| 100 | 218.5415   (s)       |~12 (m)                 | 2.7827 (s)      |
> > > | 2,000,000| 100 | 526.7188   (s)       | ~30 (m)                | out of memory          |
> > > |----------|-----|------------------|--------------------|-------------|
> > > | 50,000   | 1000 | 9.1104s    (s)      | 376.9238s   (s)          | 0.3659 (s)      |
> > > | 100,000  | 1000 | 12.7296s   (s)       | ~100 (m)       | 0.4310   (s)    |
> > > | 1,000,000| 1000 | 64.2492s   (s)       |     too slow          | 2.7827 (s)      |
> > > | 2,000,000| 1000 | 132.1232s   (s)       | too slow                  | out of memory          |
> > > |----------|-----|------------------|--------------------|-------------|
> > > | 50,000   | 10000 | 9.4384     (s)      | 376.9238s   (s)          | 0.3659 (s)      |
> > > | 100,000  | 10000 | 14.4618s   (s)       | too slow         | 0.4310   (s)    |
> > > | 1,000,000| 10000 | 67.6551s   (s)       | too slow                  | 2.7827 (s)      |
> > > | 2,000,000| 10000 | 133.4081s   (s)       | too slow                | out of memory           |
> > >
> > >
> > > As noted by the reviewer, full SW is very fast when all samples can be kept in memory. However, when the number of atoms increases to 2 million, we are no longer able to compute full SW on the GPU, even though the raw data itself can still be loaded onto the device. This supports our claim that memory constraints become a practical bottleneck in large-scale settings.
> > >
> > > For Stream-SW, the computational time increases with $n$, but the growth remains consistent with linear complexity, highlighting the scalability of our algorithm. Moreover, Stream-SW can still be computed for 2 million atoms (and beyond), since the compaction step substantially reduces the number of samples before the final computation. Interestingly, when increasing $k$ from $100$  to $1,000$, the runtime decreases. This is because a larger $k$ results in fewer compaction operations for the KKL sketches, as discussed in the paper. Increasing $k$ further to  $10,000$ leads to only a slight increase in runtime.
> > >
> > > For SW with random sampling, the reservoir-sampling-based implementation is considerably more expensive and scales poorly with both $n$ and  $k$. In particular, starting from  $n=1,000,000$ with $k=1000$ or  $n =100,000$ with $k =10,000$, the computation becomes impractical, requiring more than 6 hours (estimated) and therefore not being completed in our experiments.
> > >
> > > In conclusion, we believe that this additional runtime analysis further demonstrates the advantages of Stream-SW in streaming settings under memory constraints. We also emphasize that the reported runtimes for both Stream-SW and SW with random sampling mainly reflect the streaming computation (i.e., the compaction stage), which is currently implemented in pure Python without optimized libraries or parallelization. Therefore, there remains substantial room for further speed improvements through more efficient implementation.
> > >
> > > We will include this runtime analysis in the revised version of the paper. We hope that these additional results address the reviewer’s concerns and help clarify the practical merits of our approach. We would be happy to answer any further questions.

---

### Decision · Program_Chairs · 2026-04-30

**Decision:**

Accept (regular)

**Comment:**

This paper studies sliced Wasserstein estimation in the streaming setting and proposes Stream-SW, the first streaming estimator for sliced Wasserstein distance, built from a streaming quantile-sketch estimator for one-dimensional Wasserstein distance and extended across random projections to obtain a memory-efficient sliced OT estimator. Its main contributions are this new estimator, theoretical approximation and memory guarantees, and experiments on synthetic distributions and downstream tasks including point cloud classification, gradient flows, and change-point detection. The principal strengths are the clear novelty of the streaming setting, the solid theoretical grounding, and the practical relevance of enabling sliced Wasserstein computation under strict memory constraints. The main weaknesses are that the original submission left an important gap in runtime and efficiency analysis, had several presentation and clarity issues, and could have compared against a broader range of streaming alternatives; the rebuttal substantially addressed the first two concerns through runtime tables, clarifications of the intended regime, and promised exposition fixes, while only partially addressing the breadth of comparisons. Overall, the paper offers a novel and technically strong contribution to streaming OT that clears the bar for acceptance, and I recommend acceptance.